# NodePiece: Compositional and Parameter-Efficient Representations of Large Knowledge Graphs

**Mikhail Galkin, Etienne Denis, Jiapeng Wu and William L. Hamilton**
Mila, McGill University
Montreal, Canada
`{mikhail.galkin,deniseti,jiapeng.wu,hamilton}@mila.quebec`

## Abstract

Conventional representation learning algorithms for knowledge graphs (KG) map each entity to a unique embedding vector. Such a shallow lookup results in a linear growth of memory consumption for storing the embedding matrix and incurs high computational costs when working with real-world KGs. Drawing parallels with subword tokenization commonly used in NLP, we explore the landscape of more parameter-efficient node embedding strategies. To this end, we propose NodePiece, an anchor-based approach to learn a fixed-size entity vocabulary. In NodePiece, a vocabulary of subword/sub-entity units is constructed from anchor nodes in a graph with known relation types. Given such a fixed-size vocabulary, it is possible to bootstrap an encoding and embedding for any entity, including those unseen during training. Experiments show that NodePiece performs competitively in node classification, link prediction, and relation prediction tasks while retaining less than 10% of explicit nodes in a graph as anchors and often having 10x fewer parameters. To this end, we show that a NodePiece-enabled model outperforms existing shallow models on a large OGB WikiKG 2 graph having ~70x fewer parameters[1].

## 1 Introduction

Representation learning tasks on knowledge graphs (KGs) often require a parameterization of each unique *atom* in the graph with a vector or matrix. Traditionally, in multi-relational KGs such *atoms* constitute a set of all nodes $n \in N$ (entities) and relations (edge types) $r \in R$ (Nickel et al., 2016). Assuming parameterization with vectors, *atoms* are mapped to $d$-dimensional vectors through shallow encoders $f_n : n \to \mathbb{R}^d$ and $f_r : r \to \mathbb{R}^d$ which scale linearly to the number of nodes and edge types[2], i.e., having $O(|N|)$ space complexity of the entity embedding matrix. Albeit efficient on small conventional benchmarking datasets based on Freebase (Toutanova & Chen, 2015) (~15K nodes) and WordNet (Dettmers et al., 2018) (~40K nodes), training on larger graphs (e.g., YAGO 3-10 (Mahdisoltani et al., 2015) of 120K nodes) becomes computationally challenging. Scaling it further up to larger subsets (Hu et al., 2020; Wang et al., 2021; Safavi & Koutra, 2020) of Wikidata (Vrandecic & Krötzsch, 2014) requires a top-level GPU or a CPU cluster as done in, e.g., PyTorch-BigGraph (Lerer et al., 2019) that maintains a 78M $\times$ 200$d$ embeddings matrix in memory (we list sizes of current best performing models in Table 1).

Taking the perspective from NLP, shallow node encoding in KGs corresponds to shallow word embedding popularized with word2vec (Mikolov et al., 2013) and GloVe (Pennington et al., 2014) that learned a *vocabulary* of 400K-2M most frequent words, treating rarer ones as *out-of-vocabulary* (OOV). The OOV issue was resolved with the ability to build infinite combinations with a finite vocabulary enabled by *subword units*. Subword-powered algorithms such as fastText (Bojanowski et al., 2017), Byte-Pair Encoding (Sennrich et al., 2016), and WordPiece (Schuster & Nakajima, 2012) became a standard step in preprocessing pipelines of large language models and allowed to construct fixed-size token vocabularies, e.g., BERT (Devlin et al., 2019) contains ~30K tokens and

---

[1]The code is available on GitHub: `https://github.com/migalkin/NodePiece`
[2]We then concentrate on nodes as usually their size is orders of magnitude larger than that of edge types.

Table 1: Node embedding sizes of state-of-the-art KG embedding models compared to BERT Large. Parameters of type *float32* take 4 bytes each. FB15k-237, WN18RR, and YAGO3-10 models as reported in Sun et al. (2019), OGB WikiKG2 as in Zhang et al. (2020c), Wikidata 5M as in Wang et al. (2021), PBG Wikidata as in Lerer et al. (2019), and BERT Large as in Devlin et al. (2019).

|  | FB15k-237 | WN18RR | YAGO3-10 | OGB WikiKG2 | Wikidata 5M | PBG Wikidata | BERT Large |
|---|---|---|---|---|---|---|---|
| Vocabulary size | 15k | 40k | 120k | 2.5M | 5M | 78M | 30k |
| Embedding dim | 2000 | 1000 | 1000 | 200 | 512 | 200 | 1024 |
| GPU RAM, GB | 0.12 | 0.15 | 0.46 | 1.87 | 9.69 | 58.1 | 0.12 |

GPT-2 (Radford et al., 2019) employs ~50K tokens. Importantly, relatively small input embedding matrices enabled investing the parameters budget into more efficient encoders (Kaplan et al., 2020).

Drawing inspiration from subword embeddings in NLP, we explore how similar strategies for *tokenizing* entities in large graphs can dramatically reduce parameter complexity, increase generalization, and naturally represent new unseen entities as using the same fixed vocabulary. To do so, tokenization has to rely on *atoms* akin to subword units and not the total set of nodes.

To this end, we propose *NodePiece*, an anchor-based approach to learn a fixed-size vocabulary $V$ ($|V| \ll |N|$) of any connected multi-relational graph. In NodePiece, the set of atoms consists of anchors and all relation types that, together, allow to construct a combinatorial number of sequences from a limited atoms vocabulary. In contrast to shallow approaches, each node $n$ is first tokenized into a unique $hash(n)$ of $k$ closest anchors and $m$ immediate relations. A key element to build a node embedding is a proper encoder function $enc(n) : hash(n) \to \mathbb{R}^d$ which can be designed leveraging inductive biases of an underlying graph or downstream tasks. Therefore, the overall parameter budget is now defined by a small fixed-size vocabulary of atoms and the complexity of the encoder function.

Our experimental findings suggest that a fixed-size NodePiece vocabulary paired with a simple encoder still yields competitive results on a variety of tasks including link prediction, node classification, and relation prediction. Furthermore, anchor-based hashing enables conventional embedding models to work in the *inductive* and *out-of-sample* scenarios when unseen entities arrive at inference time, which otherwise required tailored learning mechanisms.

## 2 RELATED WORK

**Conventional KG embedding approaches.** To the best of our knowledge, all contemporary embedding algorithms (Ji et al., 2020; Ali et al., 2020) for link prediction on KGs employ shallow embedding lookups mapping each entity to a unique embedding vector thus being linear $O(|N|)$ to the total number of nodes $|N|$ and size of an embedding matrix. This holds for different embedding families, e.g., translational (Sun et al., 2019), tensor factorization (Lacroix et al., 2018), convolutional (Dettmers et al., 2018), and hyperbolic (Chami et al., 2020; Balazevic et al., 2019). The same applies to relation-aware graph neural network (GNN) encoders (Schlichtkrull et al., 2018; Vashishth et al., 2020) who still initialize each node with a learned embedding or feature vector before message passing. Furthermore, shallow encoding is also used in higher-order KG structures such as hypergraphs (Fatemi et al., 2020) and hyper-relational graphs (Rosso et al., 2020; Galkin et al., 2020). NodePiece can be used as a drop-in replacement of the embedding lookup with any of those models.

**Distillation and compression.** Several recent techniques for reducing memory footprint of embedding matrices follow successful applications of distilling large language models in NLP (Sanh et al., 2019), i.e., distillation (Wang et al., 2020; Zhu et al., 2020) into low-dimensional counterparts, and compression of trained matrices into discrete codes (Sachan, 2020). However, all of them require a full embedding matrix as input which we aim to avoid designing NodePiece.

**Vocabulary reduction in recommender systems.** Commonly, recommender systems operate on thousands of categorical features combined in sparse high-dimensional vectors. Recent approaches (Medini et al., 2021; Liang et al., 2021) employ anchor-based hashing techniques to factorize sparse feature vectors into dense embeddings. Contrary to those setups, we do not expect availability of feature vectors for arbitrary KGs and rather learn vocabulary embeddings from scratch.

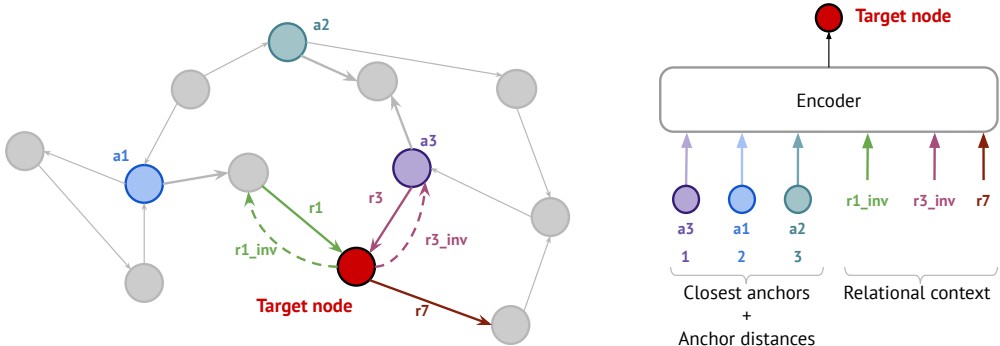

Figure 1: NodePiece tokenization strategy. Given three anchors $a_1, a_2, a_3$, a target node can be tokenized into a hash of top-$k$ closest anchors, their distances to the target node, and the relational context of outgoing relations from the target node. This hash sequence is passed through an injective encoder to obtain a unique embedding. Inverse relations are added to ensure connectivity.

**Entity descriptions and language models.** A recent line of work such as KG-BERT (Yao et al., 2019), MLMLM (Clouâtre et al., 2021), BLP (Daza et al., 2021) utilize entity descriptions passed through a language model (LM) encoder as entity embeddings suitable for link prediction. We would like to emphasize that such approaches are rather orthogonal to NodePiece. Textual features are mostly available in Wikipedia-derived KGs like Wikidata but are often missing in domain-specific graphs like social networks and product graphs. We therefore assume textual features are not available and rather learn node representations based on their spatial characteristics. Still, textual features can be easily added by concatenating NodePiece-encoded features with LM-produced features.

**Out-of-sample representation learning.** This task focuses on predictions involving previously unseen, or *out-of-sample*, entities that attach to a known KG with a few edges. These new edges are then utilized as a context to compute its embedding. Previous work (Wang et al., 2019; Hamaguchi et al., 2017; Albooyeh et al., 2020) proposed different neighborhood aggregation functions for this process or resorted to meta-learning (Chen et al., 2019; Baek et al., 2020; Zhang et al., 2020a). However, all of them follow the shallow embedding paradigm. Instead, NodePiece uses the new edges as a basis for anchor-based tokenization of new nodes in terms of an existing vocabulary.

## 3 NODEPIECE VOCABULARY CONSTRUCTION

Given a directed KG $G = (N, E, R)$ consisting of $|N|$ nodes, $|E|$ edges, and $|R|$ relation types, our task is to reduce the original vocabulary size of $|N|$ nodes to a smaller, fixed-size vocabulary of *node pieces* akin to *subword units*. In this work, we represent node pieces through *anchor* nodes $a \in A, A \subset N$, a pre-selected set of nodes in a graph following a deterministic or stochastic strategy. A full NodePiece vocabulary is then constructed from anchor nodes and relation types, i.e, $V = A + R$. Note that in order to maintain reachability of each node and balance in- and out-degrees we enrich $G$ with inverse edges with inverse relation types, such that $|R|_{inverse} = |R|_{direct}$ and $|R| = |R|_{direct} + |R|_{inverse}$. Using elements of the constructed vocabulary each node $n$ can be *tokenized* into $hash(n)$ as a sequence of $k$ closest anchors, discrete anchor distances, and a relational context of $m$ immediate relations. Then, any encoder function $enc(n) : hash(n) \rightarrow \mathbb{R}^d$ can be applied to embed the hash into a $d$-dimensional vector. An intuition of the approach is presented in Fig. 1 with each step explained in more detail below.

### 3.1 ANCHOR SELECTION

Subword tokenization algorithms such as BPE (Sennrich et al., 2016) employ deterministic strategies to create tokens and construct a vocabulary, e.g., based on frequencies of co-occurring n-grams, such that more frequent words are tokenized with fewer subword units. On graphs, such strategies might employ centrality measures like degree centrality or Personalized PageRank (Page et al., 1999). However, in our preliminary experiments, we found random anchor selection to be as effective as

centrality-based strategies. A choice for deterministic strategies might be justified when optimizing for certain task-specific topological characteristics, e.g., degree and PPR strategies indeed skew the distribution of shortest anchor distances towards smaller values thus increasing chances to find anchors in 2- or 3-hop neighborhood of any node (we provide more evidence for that in Appendix C).

## 3.2 Node Tokenization

Once the vocabulary $V = A + R$ is constructed, each node $n$ can be hashed (or *tokenized*) into a $hash(n)$ using 1) $k$ nearest anchors and their discrete distances; 2) $m$ immediate outgoing relations from the relational context of $n$. Since anchor nodes are concrete nodes in $G$, they get hashed in the same way as other non-anchor nodes.

**Anchors per node.** Given $|A|$ anchor nodes, it is impractical to use all of them for encoding each node. Instead, we select $k$ anchors per node and describe two possible strategies for that, i.e., *random* and *deterministic*. The basic random strategy uniformly samples an unordered set of $k$ anchors from $A$ yielding $\binom{|A|}{k}$ possible combinations. To avoid collisions when hashing the nodes, $|A|$ and $k$ are to be chosen according to the lower bound on possible combinations that is defined by the total number of nodes, e.g., $\binom{|A|}{k} \geq |N|$. Note that running depth-first search (DFS) to random anchors at inference time is inefficient and, therefore, $hash(n)$ of the random strategy has to be pre-computed.

On the other hand, the deterministic strategy selects an ordered sequence of $k$ nearest anchors. Hence, the anchors can be obtained via breadth-first search (BFS) in the $l$-hop neighborhood of $n$ at inference time (or pre-computed for speed reasons). However, the combinatorial bound is not applicable in this strategy and we need more discriminative signals to avoid hash collisions since nearby nodes will have similar anchors (we elaborate on the uniqueness issue in Appendix K). Such signals have to better ground anchors to the underlying graph structure, and we accomplish that using *anchor distances*[3] and *relational context* described below.

A node residing in a disconnected component is assigned with an auxiliary [DISCONNECTED] token or can be turned into an anchor. However, the majority of existing KGs are graphs with one large connected component with very few disconnected nodes, such that this effect is negligible.

**Anchor Distances.** Given a target node $n$ and an anchor $a_i$, we define anchor distance $z_{a_i} \in [0; diameter(G)]$ as an integer denoting the shortest path distance between $a_i$ and $n$ in the original graph $G$. Note that when tokenizing an anchor $a_j$ with the deterministic strategy, the nearest anchor among top-$k$ is always $a_j$ itself with distance 0. We then map each integer to a learnable $d$-dimensional vector $f_z : z_{a_i} \to \mathbb{R}^d$ akin to *relative distance encoding* scheme.

**Relational Context.** We also leverage the multi-relational nature of an underlying KG. Commonly[4], the amount of unique edge types in $G$ is orders of magnitude smaller than the total number of nodes, i.e., $|R| \ll |N|$. This fact allows to include the entire $|R|$ in the NodePiece vocabulary $V_{NP}$ and further featurize each node with a unique relational context. We construct a relational context of a node $n$ by randomly sampling a set of $m$ immediate unique outgoing relations starting from $n$, i.e., $rcon_n = \{r_j\}^m \subseteq \mathcal{N}_r(n)$ where $\mathcal{N}_r(n)$ denotes all outgoing relation types. Due to a non-uniform degree distribution, if $|\mathcal{N}_r(n)| < m$, we add auxiliary [PAD] tokens to complete $rcon_n$ to size $m$.

## 3.3 Encoding

At this step, a node $n$ is tokenized into a sequence of $k$ anchors, their $k$ respective distances, and relational context of size $m$:

$$hash(n) = \left[ \{a_i\}^k, \{z_{a_i}\}^k, \{r_j\}^m \right] \tag{1}$$

Taking anchors vectors $\mathbf{a_n}$ and relation vectors $\mathbf{r_n}$ from the learnable NodePiece vocabulary $\mathbf{V} \in \mathbb{R}^{|V| \times d}$, and anchor distances $\mathbf{z_{a_n}}$ from $\mathbf{Z} \in \mathbb{R}^{(diameter(G)+1) \times d}$, we obtain a vectorized hash:

$$hash(n) = \left[ \mathbf{a_n} + \mathbf{z_{a_n}}, \mathbf{r_n} \right] = \left[ \hat{\mathbf{a}}_\mathbf{n}, \mathbf{r_n} \right] \in \mathbb{R}^{(k+m) \times d} \tag{2}$$

---

[3]A full relational path can be mined as well but it has proven to be not scalable as each path needs to be encoded separately through a sequence encoder, e.g., GRU.

[4]As of 2021, one of the largest open KGs Wikidata contains about 100M nodes and 6K edge types

Although other operations are certainly possible, in this work, we use anchor distances as positional encodings of corresponding anchors and sum up their representations that helps to maintain the overall hash dimension of $(k + m) \times d$.

Finally, an encoder function $enc : \mathbb{R}^{(k+m) \times d} \rightarrow \mathbb{R}^d$ is applied to the vectorized hash to bootstrap an embedding of $n$. In our experiments, we probe two basic encoders: 1) MLP that takes as input a concatenated hash vector $\mathbb{R}^{1 \times (k+m)d}$ projecting it down to $\mathbb{R}^d$; 2) Transformer encoder (Vaswani et al., 2017) with average pooling that takes as input an original sequence $\mathbb{R}^{(k+m) \times d}$. While MLP is faster and better scales to graphs with more edges, Transformer is slower but requires less trainable parameters. As the two encoders were chosen to illustrate the general applicability of the whole approach, we leave a study of even more efficient and effective encoders for future work.

While the nearest-neighbor hashing function has a greater number of collisions, its non-arbitrary mapping means that it is effectively permutation invariant. We show this in Proposition 1 through the framework of Janossy pooling and permutation sampling based SGD, $\pi$-SGD (Murphy et al., 2019). A proof is provided in Appendix H.

**Proposition 1.** *The nearest-anchor encoder with $\binom{|A|}{k}$ anchors and $|m|$ subsampled relations, can be considered a $\pi$-SGD approximation of $(k + |m|)$-ary Janossy pooling with a canonical ordering induced by the anchor distances.*

Janossy pooling with $\pi$-SGD can be used to learn a permutation-invariant function from a broad class of permutation-sensitve functions such as MLPs (Murphy et al., 2019). The permutation-invariant nature of the nearest-neighbor encoding scheme combined with the lack of transductive features such as node-specific embeddings mean that NodePiece can be used for inductive learning tasks as well.

With a fixed-size vocabulary $V_{NP}$, the overall complexity and parameter budget of downstream models are largely defined by the complexity of the encoder and its inductive biases. By design, the NodePiece *smaller vocabulary - larger encoder* framework is similar to various Transformer-based language models (Qiu et al., 2020) whose vocabulary size remains rather stable with the encoder being the most important part responsible for the final performance.

## 4 EXPERIMENTS

We design the experimental program not seeking to outperform the best existing approaches but to show the versatility of NodePiece on a variety of KG-related tasks: transductive, inductive, out-of-sample link prediction, and node classification (with relation prediction results in Appendix I). With this desiderata, we formulate the following research questions: **RQ 1)** Is it necessary to map each node to a unique vector for an acceptable performance on KG tasks?; **RQ 2)** What is the effect of hashing features?; **RQ 3)** Is there an optimal number of anchors per node, after which diminishing returns hit the performance?

### 4.1 TRANSDUCTIVE LINK PREDICTION

**Setup.** We run experiments on five KGs of different sizes (Appendix A.1) varying the total number of nodes from ~15K to ~2.5M. As a baseline, we compare to RotatE (Sun et al., 2019) that remains one of state-of-the-art shallow embedding models for transductive link prediction tasks. To balance with NodePiece, RotatE operates on a graph with added inverse edges as well. We report MRR with Hits@10 in the *filtered* (Bordes et al., 2013) setting as evaluation metrics, and count parameters for all models. On larger KGs, we also compare to a smaller RotatE with a similar parameter budget.

In this task, NodePiece is equipped with a 2-layer MLP encoder. For a fair comparison, we also adopt the RotatE scoring function as a link prediction decoder. As to the NodePiece configuration, we generally keep the number of anchors below 10% of total nodes in respective graphs. We select 1k/20 for FB15k-237 (i.e., total 1000 anchors and 20 anchors per tokenized node) with 15 unique outgoing relations in the relational context; 500/50 with 4 relations for WN18RR; 7k/20 with 6 relations for CoDEx-L; 10k/20 with 5 relations for YAGO 3-10. Other hyperparameters are listed in Appendix A.

**Discussion.** Generally, the results suggest that a fixed-size NodePiece vocabulary of <10% of nodes sustains 80-90% of Hits@10 compared to 10x larger best shallow models. Some performance loss is expected due to the compositional and compressive nature of entity tokenization. On smaller graphs

Table 2: Transductive link prediction on smaller KGs. † results taken from (Sun et al., 2019). $|V|$ denotes vocabulary size (anchors + relations), #P is a total parameter count (millions). % denotes the Hits@10 ratio based on the strongest model.

| | FB15k-237 | | | | | WN18RR | | | | |
|---|---|---|---|---|---|---|---|---|---|---|
| | $|V|$ | #P (M) | MRR | H@10 | % | $|V|$ | #P (M) | MRR | H@10 | % |
| RotatE | 15k + 0.5k | 29 | $0.338^\dagger$ | $0.533^\dagger$ | 100 | 40k + 22 | 41 | $0.476^\dagger$ | $0.571^\dagger$ | 100 |
| NodePiece + RotatE | 1k + 0.5k | 3.2 | 0.256 | 0.420 | 79 | 500 + 22 | 4.4 | 0.403 | 0.515 | 90 |
| - no rel. context | 1k + 0.5k | 2 | 0.258 | 0.425 | 80 | 500 + 22 | 4.2 | 0.266 | 0.465 | 81 |
| - no distances | 1k + 0.5k | 3.2 | 0.254 | 0.421 | 79 | 500 + 22 | 4.4 | 0.391 | 0.510 | 89 |
| - no anchors, rels only | 0 + 0.5k | 1.4 | 0.204 | 0.355 | 67 | 0 + 22 | 0.3 | 0.011 | 0.019 | 0.3 |

Table 3: Transductive link prediction on bigger KGs. The same denotation as in Table 2. Second RotatE has a similar parameter budget as a NodePiece-based model.

| | CoDEx-L | | | | | YAGO 3-10 | | | | |
|---|---|---|---|---|---|---|---|---|---|---|
| | $|V|$ | #P (M) | MRR | H@10 | % | $|V|$ | #P (M) | MRR | H@10 | % |
| RotatE (500d) | 77k + 138 | 77 | 0.258 | 0.387 | 100 | 123k + 74 | 123 | $0.495^\dagger$ | $0.670^\dagger$ | 100 |
| RotatE (20d) | 77k + 138 | 3.8 | 0.196 | 0.322 | 83 | 123k + 74 | 4.8 | 0.121 | 0.262 | 39 |
| NodePiece + RotatE | 7k + 138 | 3.6 | 0.190 | 0.313 | 81 | 10k + 74 | 4.1 | 0.247 | 0.488 | 73 |
| - no rel. context | 7k + 138 | 3.1 | 0.201 | 0.332 | 86 | 10k + 74 | 3.7 | 0.249 | 0.482 | 72 |
| - no distances | 7k + 138 | 3.6 | 0.179 | 0.302 | 78 | 10k + 74 | 4.1 | 0.250 | 0.491 | 73 |
| - no anchors, rels only | 0 + 138 | 0.6 | 0.063 | 0.121 | 31 | 0 + 74 | 0.5 | 0.025 | 0.041 | 6 |

(Table 2), parameter saving might not be well pronounced due to the overall small number of nodes to embed. Still, taking even as few as 500 nodes as anchors on WN18RR retains 90% of the best model performance. On bigger graphs (Table 3), parameter efficiency is more pronounced, i.e., on YAGO 3-10, a RotatE model of comparable size is 20 Hits@10 points worse than a NodePiece-based one. This observation can be attributed to the fact the shrinking shallow models results in shrinking the embedding dimension of each node (20d for RotatE) which is inefficient on small parameter budgets. In contrast, a small fixed-size vocabulary allows for larger anchor embedding dimensions (100d for NodePiece with RotatE) since most of the parameter budget is defined by the encoder.

We further study the effect of different anchor selection combinations (Fig. 2). On WN18RR, fewer anchors with fewer anchors per node ($|A|/k$) yield relatively low accuracy but starting from 50/20 (~0.1% of 40k nodes in the graph) the Hits@10 performance starts to saturate. On FB15k-237, as few as 25 anchors already exhibit the signs of saturation where a further increase to 500 or 1000 anchors only marginally improves the performance. We hypothesize such a difference can be explained by graph density, e.g., WN18RR is a sparse graph with a diameter of 23 and average anchor distance of 6 hops; while FB15k-237 is a denser graph with an average anchor distance of 2-3. Hence, on a sparse graph with longer distances, it takes more anchors to properly encode a node.

However, more precise predictions (e.g., Hits@1) reflected in the MRR metric (see Appendix E) still remain a challenging task for small vocabulary NodePiece setups, and bigger $|A|/k$ combinations alleviate this issue. We also observe that diminishing returns, which make further vocabulary increase less rewarding, start to appear from anchor set sizes of ~1% of total nodes.

**Ablations.** In the ablation study, we measure the impact of relational context and anchor distances on link prediction (Table 2). Removing relational context and anchor distances does not tangibly affect the denser FB15k-237 data but does impair the accuracy on the sparser WN18RR. Pushing vocabulary sizes to the limit, we also investigate NodePiece behavior in the absence of anchors at all, i.e., when hashes are defined only by the relational context of size $m$. Interestingly, this still yields fair performance on FB15k-237 with just 7 points Hits@10 drop, but drops to zero the WN18RR performance. The fact that node embeddings might not be at all necessary but relations are more important supports the recent findings of Teru et al. (2020) that relies only on relations seen in a small subgraph around a target node. However, at this point, it seems to be a virtue of graphs with a diverse set of unique relations. That is, FB15k-237 has 20x more unique relations than WN18RR and resulting hashes have more diverse combinations of relations which lead to more discriminative node representations. Additionally, we visualize anchor embedding projections in Appendix D.

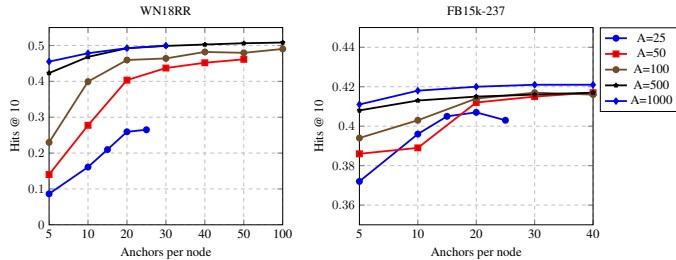

Figure 2: Combinations of total anchors $A$ and anchors per node. Denser FB15k-237 saturates faster on smaller $A$ while sparse WN18RR saturates at around 500 anchors.

### 4.1.1 OGB WIKIKG 2

To measure the benefits of NodePiece on large-scale KGs, we run a link prediction experiment on OGB WikiKG 2 (Hu et al., 2020), a subset of Wikidata that consists of 2.5M nodes and 16M edges. NodePiece is configured to sample a vocabulary 20K anchor nodes ($< 1\%$ of total nodes) where each node is represented with $k = 20$ nearest anchors and a relational context of size $m = 12$, and we use a 2-layer MLP as a hash encoder (other hyperparameters as in Appendix A). Generally, such a NodePiece configuration can be paired with any link prediction decoder and we chose a non-parametric AutoSF (Zhang et al., 2020b) as one of the strongest decoders on this graph. Overall, the NodePiece + AutoSF model has only 6.9M parameters, about $70\times$ smaller than top shallow models. Compared to the best reported shallow approaches,

Table 4: Test MRR and parameter budget on OGB WikiKG 2.

| Model | #Params | MRR |
|---|---|---|
| NP + AutoSF | 6.9M | $0.570_{\pm0.003}$ |
| - rel. context | 5.9M | $0.592_{\pm0.003}$ |
| - anc. dists | 6.9M | $0.570_{\pm0.004}$ |
| - no anchors | 1.3M | $0.476_{\pm0.001}$ |
| AutoSF | 500M | $0.546_{\pm0.005}$ |
| PairRE | 500M | $0.521_{\pm0.003}$ |
| RotatE | 1250M | $0.433_{\pm0.002}$ |
| TransE | 1250M | $0.426_{\pm0.003}$ |

the NodePiece-enabled model exhibits (cf. Table 4, averaged over 10 seeds) even better performance achieved with a orders of magnitude smaller parameter budget. We believe this result shows the effectiveness of NodePiece on large KGs with a dramatic parameter size reduction without significant performance losses. Ablations report the duality of a relational context, i.e., removing it from hashes leads to even higher MRR scores. On the other hand, the relational context alone with 0 learnable anchors still yields considerably better results than $1000\times$ larger shallow models RotatE and TransE.

### 4.2 INDUCTIVE LINK PREDICTION

We conduct a set of experiments on the inductive link prediction benchmark introduced by Teru et al. (2020) to measure the performance of NodePiece features in the extreme case when anchor nodes are not available and only relational context can be used to compose entity representations.

**Setup.** The unique feature of this benchmark compared to other evaluated tasks is that training and inference graphs are disjoint, i.e., inference at validation and test time is performed on a completely new graph comprised of new entities, and link prediction involves only entities unseen during training. As inference graphs are disconnected from training ones, learning anchors from the training graph is useless, so node hashes are built only using the $m$-sized relational context. On top of the obtained NodePiece features we then employ a relational message passing GNN, CompGCN (Vashishth et al., 2020), with RotatE (Sun et al., 2019) as a scoring function for triples. More details on the setup and best hyperparameters for NodePiece are presented in Appendix J.

**Baselines**. We compare NodePiece + CompGCN with two families of models applicable in the inductive setting, i.e., rule-based methods, Neural LP (Yang et al., 2017), DRUM (Sadeghian et al., 2019), RuleN (Meilicke et al., 2018), and GNNs: GraIL (Teru et al., 2020) and recently proposed Neural Bellman-Ford Nets (NBFNet) (Zhu et al., 2021).

**Discussion.** Generally, the results confirm the trend identified previously: relation-only features are strong performers in dense relation-rich graphs. NodePiece features paired with CompGCN significantly improve over path-based methods where performance gap might reach 37 absolute Hits@10 points, e.g., in FB15k-237 V1 and NELL-995 V1. Comparing to GNNs, NodePiece +

Table 5: Inductive link prediction results, Hits@10. Best results are in **bold**, second best are underlined. † results taken from Teru et al. (2020). NBFNet results taken from Zhu et al. (2021).

| Class | Method | FB15k-237 | | | | WN18RR | | | | NELL-995 | | | |
|-------|--------|------|------|------|------|------|------|------|------|------|------|------|------|
| | | V1 | V2 | V3 | V4 | V1 | V2 | V3 | V4 | V1 | V2 | V3 | V4 |
| Path | Neural LP † | 0.529 | 0.589 | 0.529 | 0.559 | 0.744 | 0.689 | 0.462 | 0.671 | 0.408 | 0.787 | 0.827 | 0.806 |
| | DRUM † | 0.529 | 0.587 | 0.529 | 0.559 | 0.744 | 0.689 | 0.462 | 0.671 | 0.194 | 0.786 | 0.827 | 0.806 |
| | RuleN † | 0.498 | 0.778 | 0.877 | 0.856 | 0.809 | 0.782 | 0.534 | 0.716 | 0.535 | 0.818 | 0.773 | 0.614 |
| GNN | GraIL † | 0.642 | 0.818 | 0.828 | 0.893 | 0.825 | 0.787 | 0.584 | 0.734 | 0.595 | **0.933** | 0.914 | 0.732 |
| | NBFNet | 0.834 | **0.949** | **0.951** | **0.960** | **0.948** | **0.905** | **0.893** | **0.890** | - | - | - | - |
| | NP + CompGCN | **0.873** | 0.939 | 0.944 | 0.949 | 0.830 | 0.886 | 0.785 | 0.807 | **0.890** | 0.901 | **0.936** | **0.893** |

Table 6: Node classification results. $|V|$ denotes vocabulary size (anchors + relations), #P is a total parameter count (millions).

| | $|V|$ | #P (M) | WD50K (5% labeled) | | | WD50K (10% labeled) | | |
|---|-------|--------|---------|---------|----------|---------|---------|----------|
| | | | ROC-AUC | PRC-AUC | Hard Acc | ROC-AUC | PRC-AUC | Hard Acc |
| MLP | 46k + 1k | 4.1 | 0.503 | 0.016 | 0.001 | 0.510 | 0.017 | 0.002 |
| CompGCN | 46k + 1k | 4.4 | 0.836 | 0.280 | 0.176 | 0.834 | 0.265 | 0.161 |
| NodePiece + GNN | 50 + 1k | 0.75 | 0.981 | 0.443 | 0.513 | 0.981 | 0.450 | 0.516 |
| - no rel. context | 50 + 1k | 0.64 | 0.982 | 0.446 | 0.534 | 0.982 | 0.449 | 0.530 |
| - no distances | 50 + 1k | 0.74 | 0.981 | 0.448 | 0.516 | 0.981 | 0.448 | 0.513 |
| - no anchors, rels only | 0 + 1k | 0.54 | 0.984 | 0.453 | 0.532 | 0.984 | 0.456 | 0.533 |

CompGCN outperforms GraIL by a large margin in all (except one) experiments and is competitive to NBFNet on relation-rich FB15k-237 splits. As expected, NodePiece features are less efficient on sparse graphs (like WN18RR with few unique relations) but still outperform topology-based GraIL.

## 4.3 NODE CLASSIFICATION

**Setup.** Due to the lack of established node classification datasets on multi-relational KGs, we design a multi-class multi-label task based on a triple version of a recent WD50K (Galkin et al., 2020) extracted from Wikidata. The pre-processing steps are described in Appendix F, and the final graph consists of 46K nodes and 222K edges. The task belongs to the family of transductive (the whole graph is seen during training) semi-supervised (only a fraction of nodes are labeled) problems, where labels are 465 classes as seen in Wikidata. In a semi-supervised mode, we test the models on a graph with 5% and 10% of labeled nodes. Node features have to be learned as node embeddings.

As baselines, we compare to a 2-layer MLP and CompGCN (Vashishth et al., 2020) in a full-batch mode which is one of the strongest GNN encoders for multi-relational KGs. Both baselines learn a full entity vocabulary. We report ROC-AUC, PRC-AUC, and Hard Accuracy metrics as commonly done in standard graph benchmarks like OGB (Hu et al., 2020). For PRC-AUC and Hard Accuracy, we binarize predicted logits using a threshold of 0.5. Hard Accuracy corresponds to the exact match of a predicted sparse 465-dimensional vector to a sparse 465-dimensional labels vector.

NodePiece is configured to have only 50 anchors and use 10 nearest anchors per node with 5 unique relations in the relational context. The dimensionality of anchors and relations is the same as in the baseline CompGCN. Each epoch, we first materialize all entity embeddings through the NodePiece encoder and then send the materialized matrix to CompGCN with the class predictor.

**Discussion.** Surprisingly, ~1000x vocabulary reduction ratio (50 anchors against 46k for shallow models) greatly outperforms the baselines (Table 6). MLP, as expected, is not able to cope with the task producing random predictions. CompGCN, in turn, outperforms MLP demonstrating non-random outputs as seen by the ROC-AUC score of 0.836 and higher PRC-AUC and Hard Accuracy metrics. Still, a NodePiece-equipped CompGCN with 50 anchors reaches even higher ROC-AUC of 0.98 with considerable improvements along other metrics, i.e., +16-19 PRC-AUC points and 3x boost along the hardest accuracy metric. We attribute such a noticeable performance difference to better generalization capabilities of the NodePiece model. That is, a generalization gap between training and validation metrics of the NodePiece + CompGCN is much smaller compared to the baselines who overfit rather heavily ( cf. the training curves in the Appendix G). The effect remains after increasing

Table 7: Out-of-sample link prediction. † results are taken from (Albooyeh et al., 2020). $|V|$ denotes vocabulary size (anchors + relations), #P is a total parameter count (millions).

| | oFB15k-237 | | | | | oYAGO 3-10 (117k) | | | | |
|---|---|---|---|---|---|---|---|---|---|---|
| | $|V|$ | #P (M) | MRR | H@10 | % | $|V|$ | #P (M) | MRR | H@10 | % |
| oDistMult-ERAvg | 11k + 0.5k | 2.4 | 0.256† | 0.420† | 100 | 117k + 74 | 23.4 | OOM | OOM | - |
| NodePiece + DistMult | 1k + 0.5k | 1 | 0.206 | 0.372 | 88 | 10k + 74 | 2.7 | 0.133 | 0.261 | 100 |
| - no rel. context | 1k + 0.5k | 1 | 0.173 | 0.329 | 78 | 10k + 74 | 2.7 | 0.125 | 0.245 | 94 |
| - no distances | 1k + 0.5k | 1 | 0.208 | 0.372 | 88 | 10k + 74 | 2.7 | 0.133 | 0.260 | 99 |
| - no anchors, rels only | 0 + 0.5k | 0.8 | 0.069 | 0.127 | 30 | 0 + 74 | 0.7 | 0.015 | 0.017 | 6 |

the number of labeled nodes to 10%. Even with 50 anchors, the overall performance is saturated as the further increase of the vocabulary size did not bring any improvements.

**Ablations.** We probe setups where NodePiece hashes use only anchors or only relational context, and find they both deliver a similar performance. Following the previous experiments on dense graphs with lots of unique relations, it appears that node classification can be performed rather accurately based only on the node relational context which is captured by NodePiece hashes.

## 4.4 OUT-OF-SAMPLE LINK PREDICTION

**Setup.** In the out-of-sample setup, validation and test splits contain unseen entities that arrive with a few edges connected to the seen nodes. For this experiment, we use the out-of-sample FB15k-237 split (oFB15k-237) as designed in Albooyeh et al. (2020). We do not employ their version of WN18RR as the split contains too many disconnected entities and components in the train graph. Instead, using the authors script, we sample a much bigger out-of-sample version of YAGO 3-10.

As a baseline, we compare to oDistMult (Albooyeh et al., 2020) which aggregates embeddings of all seen neighboring nodes around the unseen one (akin to 1-layer message passing with mean aggregator). We adopt the same evaluation protocol - given an unseen node with its connecting edges, we mask one of the edges and predict its tail or head using the rest of the edges, repeating this procedure for each edge. We report filtered MRR and Hits@10 as main metrics.

NodePiece enables traditional transductive-only models to perform inductive inference as both seen and unseen nodes are tokenized using the same vocabulary. For a smaller oFB15k-237 the NodePiece vocabulary has 1k/20 configuration with 15 relations, while in a bigger oYAGO 3-10 we use 10k/20 with 5 relations. For this task, we apply a transformer encoder instead of MLP. For a fair comparison, we use DistMult as a scoring function as well.

**Discussion.** The results in Table 7 show that a simple NodePiece-based model retains ~90% of the baseline performance on oFB15k-237, but achieved faster and computationally inexpensive compared to oDistMult. Moreover, while oDistMult is tailored specifically for the out-of-sample task, we did not do any task-specific modifications to the NodePiece-enabled model as it is inductive by design. Furthermore, oDistMult is not able to scale to a bigger oYAGO 3-10 on a 256 GB RAM machine due to the out of memory crash. Conversely, a NodePiece-equipped model has the same computational requirements as in other tasks and converges rather quickly (40 epochs). Performed ablations underline the importance of having both anchors and relational context for tokenizing unseen entities. We elaborate more on possible inference strategies for transductive and inductive tasks in Appendix A.6.

## 5 CONCLUSION

In this paper, we have introduced NodePiece, a compositional approach for representing nodes in multi-relational graphs with a fixed-size vocabulary. Similar to subword units, NodePiece allows to tokenize every node as a combination of anchors and relations where the number of anchors can be 10–100× smaller than the total number of nodes. We show that in some tasks, node embeddings are not even necessary for getting an acceptable accuracy thanks to a rich set of relation types. Moreover, NodePiece is inductive by design and is able to tokenize unseen entities and perform downstream prediction tasks in the same fashion as on seen ones.

**Reproducibility Statement.** The source code is openly available on GitHub. All hyperparameters and implementation details are presented in Appendix A. Information on the used datasets is presented in Table 8 and we provide more details on dataset construction for node classification and out-of-sample link prediction tasks in Appendix F. The proof for Proposition 1 is given in the Appendix H.

**Ethics Statement.** As NodePiece is a general graph representation learning method, we do not foresee immediate ethical consequences pertaining to the method itself.

**Acknowledgements.** The authors would like to thank Koustuv Sinha, Gaurav Maheshwari, and Priyansh Trivedi for insightful and valuable discussions at earlier stages of this work. We also thank anonymous reviewers for the helpful comments. This work is partially supported by the Canada CIFAR AI Chair Program and Samsung AI grant (held at Mila). We thank Mila and Compute Canada for access to computational resources.

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

# A  IMPLEMENTATION & HYPERPARAMETERS

Table 8: Dataset statistics. LP - link prediction, RP - relation prediction, NC - node classification, OOS - out-of-sample. In OOS-LP, *Nodes* also shows the amount of unseen nodes in validation/test.

| Dataset | Task | Nodes | Relations | Edges | Train | Validation | Test |
|---|---|---|---|---|---|---|---|
| FB15k-237 (Toutanova & Chen, 2015) | LP, RP | 14,505 | 237 | 310,079 | 272,115 | 17,526 | 20,438 |
| WN18RR (Dettmers et al., 2018) | LP, RP | 40,559 | 11 | 92,583 | 86,835 | 2824 | 2924 |
| CoDEx-Large (Safavi & Koutra, 2020) | LP | 77,951 | 69 | 612,437 | 551,193 | 30,622 | 30,622 |
| YAGO 3-10 (Mahdisoltani et al., 2015) | LP, RP | 123,143 | 37 | 1,089,000 | 1,079,040 | 4978 | 4982 |
| OGB WikiKG 2 (Hu et al., 2020) | LP | 2,500,604 | 535 | 17,137,181 | 16,109,182 | 429,456 | 598,543 |
| WD50K | NC | 46,164 | 526 | 222,563 | 4600 (N) | 4600 (N) | 4600 (N) |
| oFB15k-237 (Albooyeh et al., 2020) | OOS-LP | 11k/1395/1395 | 234 | 292,173 | 193,490 | 44,601 | 54,082 |
| oYAGO 3-10 | OOS-LP | 117k/2960/2959 | 37 | 1,086,416 | 988,124 | 47,112 | 51,180 |

Table 9: Inductive relation prediction dataset statistics. Facts denote the size of the input graph while queries denote the triples to be predicted. Training sets contain all queries as facts. Note that in validation and test we receive a new graph disjoint from the training one, and queries are sent against this new inference graph (hence the number of entities and facts for validation and test is the same).

| Dataset | | Relations | Train | | | Validation | | | Test | | |
|---|---|---|---|---|---|---|---|---|---|---|---|
| | | | Entity | Query | Facts | Entity | Query | Fact | Entity | Query | Facts |
| FB15k-237 | v1 | 183 | 2,000 | 4,245 | 4,245 | 1,500 | 206 | 1,993 | 1,500 | 205 | 1,993 |
| | v2 | 203 | 3,000 | 9,739 | 9,739 | 2,000 | 469 | 4,145 | 2,000 | 478 | 4,145 |
| | v3 | 218 | 4,000 | 17,986 | 17,986 | 3,000 | 866 | 7,406 | 3,000 | 865 | 7,406 |
| | v4 | 222 | 5,000 | 27,203 | 27,203 | 3,500 | 1,416 | 11,714 | 3,500 | 1,424 | 11,714 |
| WN18RR | v1 | 9 | 2,746 | 5,410 | 5,410 | 922 | 185 | 1,618 | 922 | 188 | 1,618 |
| | v2 | 10 | 6,954 | 15,262 | 15,262 | 2,923 | 411 | 4,011 | 2,923 | 441 | 4,011 |
| | v3 | 11 | 12,078 | 25,901 | 25,901 | 5,084 | 538 | 6,327 | 5,084 | 605 | 6,327 |
| | v4 | 9 | 3,861 | 7,940 | 7,940 | 7,208 | 1,394 | 12,334 | 7,208 | 1,429 | 12,334 |
| NELL-995 | v1 | 14 | 3,103 | 4,687 | 4,687 | 225 | 101 | 833 | 225 | 100 | 833 |
| | v2 | 88 | 2,564 | 8,219 | 8,219 | 4,937 | 459 | 4,586 | 4,937 | 476 | 4,586 |
| | v3 | 142 | 4,647 | 16,393 | 16,393 | 4,921 | 811 | 8,048 | 4,921 | 809 | 8,048 |
| | v4 | 77 | 2,092 | 7,546 | 7,546 | 3,294 | 716 | 7,073 | 3,294 | 731 | 7,073 |

NodePiece is implemented in Python using *igraph* library (licensed under GNU GPL 2) for computing centrality measures and perform basic tokenization. Downstream tasks employ NodePiece in conjunction with PyTorch (Paszke et al., 2019) (BSD-style license), PyKEEN (Ali et al., 2021) (MIT License), and PyTorch-Geometric (Fey & Lenssen, 2019) (MIT License). We ran experiments on a machine with one RTX 8000 GPU and 64 GB RAM. The OGB WikiKG 2 experiments were executed on a single Tesla V100 16 GB VRAM and 64 GB RAM. All used datasets are available under open licenses.

For all downstream tasks and datasets we employ the deterministic anchor selection strategy where 40% of the total number of anchors $|A|$ are nodes with top PPR scores, 40% are top degree nodes, and remaining 20% are selected randomly. All anchor sets are non-overlapping and disjoint, i.e., if some top degree nodes have already been selected with the PPR policy, they will be skipped in favor of next nodes in the sorted list. The choice for this strategy is motivated in Appendix C.

## A.1  DATASETS

Details on the datasets for transductive link prediction, out-of-sample link prediction, relation prediction and node classification are collected in Table 8. The inductive link prediction benchmark introduced by Teru et al. (2020) includes 3 graphs, FB15k-237, WN18RR, and NELL-995, each has 4 different splits that vary in the number of unique relations, number of nodes and triples at training and inference time. Full dataset statistics is provided in Table 9. FB15k-237 and most splits of NELL-995 can be considered as relation-rich graphs while WN18RR is a sparse graph with few relation types.

## A.2 TRANSDUCTIVE LINK PREDICTION

The optimizer is Adam for all experiments. As RotatE is a scoring function in the complex space, the reported embedding dimensions are a sum of real and imaginary dimensions, e.g., 1000d means that both real and imaginary vectors are 500d.

Table 10: NodePiece hyperparameters for transductive link prediction experiments

| Parameter | FB15k-237 | WN18RR | CoDEx-L | YAGO 3-10 | OGB WikiKG 2 |
|---|---|---|---|---|---|
| # Anchors, $|A|$ | 1000 | 500 | 7000 | 10000 | 20000 |
| # Anchors per node, $k$ | 20 | 50 | 20 | 20 | 20 |
| Relational context, $m$ | 15 | 4 | 6 | 5 | 12 |
| Vocabulary dim, $d$ | 200 | 200 | 200 | 200 | 200 |
| Batch size | 512 | 512 | 256 | 512 | 512 |
| Learning rate | 0.0005 | 0.0005 | 0.0005 | 0.00025 | 0.0001 |
| Epochs | 400 | 600 | 120 | 600 | 300k (steps) |
| Encoder type | MLP | MLP | MLP | MLP | MLP |
| Encoder dim | 400 | 400 | 400 | 400 | 400 |
| Encoder layers | 2 | 2 | 2 | 2 | 2 |
| Encoder dropout | 0.1 | 0.1 | 0.1 | 0.1 | 0.1 |
| Loss function | BCE | NSSAL | BCE | NSSAL | NSSAL |
| Margin | - | 15 | - | 50 | 50 |
| # Negative samples | - | 20 | - | 10 | 128 |
| Label smoothing | 0.4 | - | 0.3 | - | - |
| Training time, hours | 7 | 5.5 | 26 | 23 | 11 |

Table 11: RotatE hyperparameters for transductive link prediction experiments. CoDEx-L and YAGO 3-10 also list the hyperparameters (after the symbol / ) for smaller models (reported in Table 3) of the same parameter budget as NodePiece

| Parameter | FB15k-237 | WN18RR | CoDEx-L | YAGO 3-10 |
|---|---|---|---|---|
| Embedding dim, $d$ | 2000 | 1000 | 1000 / 50 | 1000 / 40 |
| Batch size | 1024 | 512 | 512 / 512 | 1024 / 512 |
| Loss function | NSSAL | NSSAL | NSSAL | NSSAL |
| Margin | 9 | 6 | 25 / 9 | 24 / 15 |
| # Negative samples | 256 | 1024 | 100 / 100 | 400 / 100 |

## A.3 RELATION PREDICTION

Configurations (Table 12) for the compared models are almost identical to those of the transductive link prediction experiment. We mostly reduce the number of epochs and negative samples as models converge faster on this task.

## A.4 NODE CLASSIFICATION

In this experiment (Table 13), NodePiece is used at the initial step to bootstrap a node embeddings matrix which is then sent to the CompGCN graph encoder. In contrast, CompGCN and MLP baselines use directly a trained node embedding matrix as their initial input.

## A.5 OUT-OF-SAMPLE LINK PREDICTION

The set of NodePiece hyperparameters (Table 14) is similar to the set of the transductive experiments except the scoring function (DistMult), encoder function (Transformer), and number of epochs as the model converges faster. We do not provide a setup for the baseline oDistMult on oYAGO 3-10 as the model was not able to pre-process the dataset on a machine with 256 GB RAM. Reported training

Table 12: Hyperparameters for relation prediction experiments. The content is largely identical to Table 10, only changed parameters are listed

| Parameter | NodePiece + RotatE | | | RotatE | | |
|---|---|---|---|---|---|---|
| | FB15k-237 | WN18RR | YAGO 3-10 | FB15k-237 | WN18RR | YAGO 3-10 |
| Batch size | 512 | 512 | 512 | 512 | 512 | 512 |
| Epochs | 20 | 150 | 7 | 150 | 150 | 150 |
| Loss function | NSSAL | NSSAL | NSSAL | NSSAL | NSSAL | NSSAL |
| Margin | 15 | 12 | 25 | 9 | 3 | 5 |
| # Negative samples | 20 | 20 | 20 | 20 | 20 | 20 |
| Training time, min | 25 | 30 | 25 | 28 | 10 | 57 |

Table 13: Hyperparameters for node classification experiments

| Parameter | NodePiece + CompGCN | CompGCN | MLP |
|---|---|---|---|
| # Anchors, $|A|$ | 50 | - | - |
| # Anchors per node, $k$ | 10 | - | - |
| Relational context, $m$ | 5 | - | - |
| Vocabulary dim, $d$ | 100 | 100 | 100 |
| Batch size | 512 | 512 | 512 |
| Learning rate | 0.001 | 0.001 | 0.001 |
| Epochs | 4000 | 4000 | 4000 |
| NodePiece encoder | MLP | - | - |
| NodePiece encoder dim | 200 | - | - |
| NodePiece encoder layers | 2 | - | - |
| NodePiece encoder dropout | 0.1 | - | - |
| GNN (MLP) layers | 3 | 3 | 3 |
| GNN (MLP) dropout | 0.5 | 0.5 | 0.5 |
| Loss function | BCE | BCE | BCE |
| Label smoothing | 0.1 | 0.1 | 0.1 |
| Training time, hours | 14 | 22 | 6 |

times for NodePiece models exclude evaluation. Training times of the baseline oDistMult were not reported by its authors.

## A.6 DEPLOYMENT IN REAL-WORLD DYNAMIC KNOWLEDGE GRAPHS

NodePiece, on account of its compositional representation, can be applied to dynamic real-world knowledge graphs where nodes are added and removed over time. That is, training on a graph snapshot we can obtain the embeddings of new nodes without re-computing and updating representations of every other node in the graph. This is valuable in settings where there are latency requirements such as many online services. For example, if a user creates an account on a social media service and begins liking content (represented as a "like" edge in the social network graph between the user and the content), it would be desirable to make future content recommendations rapidly reflect this new data without waiting for the next batched retraining to update the users embedding. In order to reduce latency the entire embedding matrix can be materialized and cached ahead of time, updating embeddings as new nodes and edges are added. The parameter efficiency and compositionality of NodePeice means that for large real-world graphs NodePiece subsumes what would before have been a complex system of a large-scale embedding framework like Pytorch-BigGraph (Lerer et al., 2019), an OOV embedding method (e.g,. ERAvg) and a shallow embedding method (e.g., RotatE).

Table 14: Hyperparameters for out-of-sample link prediction experiments. The content is largely identical to Table 10, only changed parameters are listed

| | NodePiece + DistMult | | oDistMult |
| Parameter | oFB15k-237 | oYAGO 3-10 | oFB15k-237 |
|---|---|---|---|
| # Anchors, $|A|$ | 1000 | 10000 | - |
| # Anchors per node, $k$ | 20 | 20 | - |
| Relational context, $m$ | 15 | 5 | - |
| Vocabulary dim, $d$ | 200 | 200 | 200 |
| Batch size | 256 | 256 | 1000 |
| Learning rate | 0.0005 | 0.0005 | 0.01 |
| Epochs | 40 | 40 | 1000 |
| NodePiece encoder | Transformer | Transformer | - |
| NodePiece encoder dim | 512 | 512 | - |
| NodePiece encoder layers | 2 | 2 | - |
| NodePiece encoder dropout | 0.1 | 0.1 | - |
| Loss function | Softplus | Softplus | Softplus |
| # Negative samples | 5 | 5 | 1 |
| Training time, hours | 2 | 8 | - |

## B  LIMITATIONS AND FUTURE WORK

Our experimental results demonstrate the promise of using NodePeice to significantly reduce the parameter complexity of node embeddings. While it is difficult to prove, we also hypothesize that the parameters required by NodePeice to maintain the same level of performance (as the graph scales) increase sublinearly according to the size of the graph. The intuition for this is twofold. First, the number of unique anchor combinations of size $k$ that can be encoded increases according to $\binom{|A|}{k}$ (i.e. $\mathcal{O}(|A|^k)$) if randomly sampled — if the sampling is done via nearest neighbor anchor selection then the number of unique permutations is expected to increase polynomially. Second, increasing the size of the graph will only require sublinear increase in the number of anchors in order to maintain the same average node-anchor distance. Although proving causality is difficult, we believe that maintaining hashing uniqueness and node-anchor distances stable will be sufficient to maintain equivalent performance.

## C  ANCHOR SELECTION STRATEGIES

Here, we provide more details as to anchor configurations ($k$ nearest from total $A$ anchors) and anchor distances. Recall that there exist several ways to select the total set of anchors $A$ as stated in Section 3.1, i.e., random or centrality-measure based. Then, $k$ anchors per node can be chosen either as $k$ nearest (default NodePiece mode) or $k$ random anchors. Figure 3 depicts the effect of those strategies on the distribution of anchor distances (number of hops between a target node and its anchors). We use the configurations used in the main experiments, i.e., 1000 anchors and 20 anchors per node for FB15k-237, and 500 anchors with 50 anchors per node on WN18RR.

First, we observe that PPR, degree, and mixed (40% PPR, 40% degree, 20% random) strategies generally skew the distribution towards smaller anchor distances compared to random strategies. This fact supports the hypothesis that deterministic strategies improve the chances to find an anchor in a closer $l$-hop neighborhood of a target node. Second, varying the way of selecting $k$ anchors per node between nearest (left column) and random (right column), we also observe the skew of a distribution of anchor distances.

Next, we fix the anchor selection strategy to the *mix*, fix the number of anchors per node (50 for WN18RR and 20 for FB15k-237), and vary a total number of anchors $A$ (50 to 1000 for WN18RR and 20 to 1000 for FB15k-237) along with the method of sampling $k$ anchors per node, i.e., nearest and random. Figure 4 shows that increasing the total number of anchors together with $k$ nearest

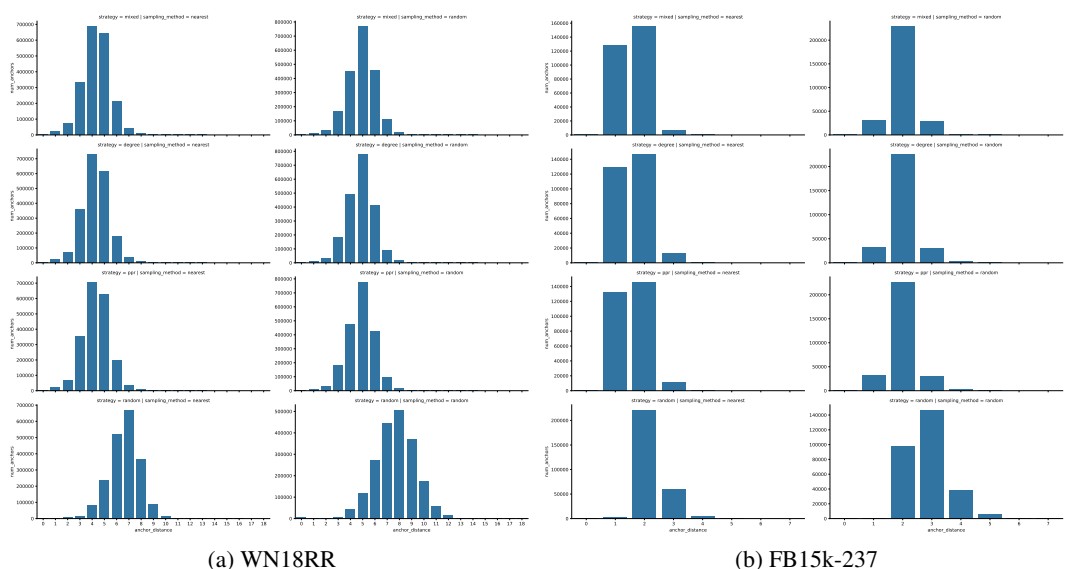

(a) WN18RR        (b) FB15k-237

Figure 3: Distribution of anchor distances under various anchor selection strategies. Top-bottom: mixed, degree-based, PPR-based, random. For each dataset, left: selecting $k$ nearest anchors, right: $k$ random anchors. (a) Selecting a fixed 500/50 configuration on WN18RR; (b) Selecting a fixed 1000/20 configuration on FB15k237. Generally, all strategies except random ones skew the distributions towards nearest anchors.

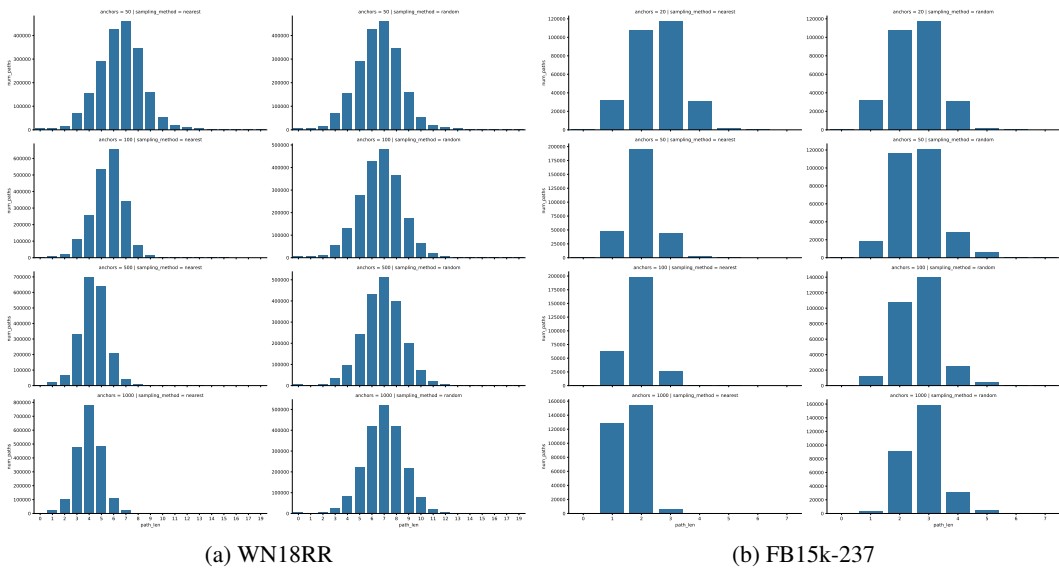

(a) WN18RR        (b) FB15k-237

Figure 4: Distribution of anchor distances under the fixed *mix* anchor selection strategy when varying the total number of anchors $A$ (50–1000 for WN18RR, 20–1000 for Fb15k-237). For each dataset, left column - $k$ nearest anchors, right - $k$ random anchors. On both graphs, increase in $A$ with the nearest anchors always leads to shorter anchor distances.

anchors again skews the distribution of anchor distances towards smaller values and, hence, to higher probabilities of finding anchors in a closer neighborhood of a target node.

We would recommend using centrality-based strategies to select $A$ with $k$ *nearest* anchors per node if anchor distances and probability of finding anchors in a closer neighborhood are of higher importance.

Finally, we fix the anchor selection strategy as *mix*, obtain *nearest* anchors per node, and under this setup study average anchor distances varying $k$ - the number of anchors per node in various combinations of total anchors $A$. The results presented on Figure 5 suggest that sparser graphs (like WN18RR) benefit more from increasing the number of anchors $A$, i.e., the delta between distances is much larger than that of dense FB15k-237. The difference in distances on Figure 5 might also explain the performance on Figure 8, i.e., generally, smaller $A/k$ configurations like 25/5 are inferior on sparser graphs but perform competitively on denser ones.

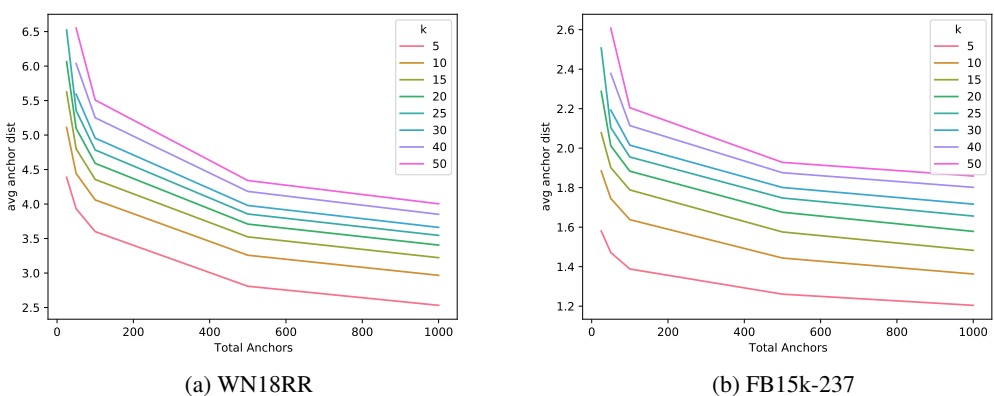

(a) WN18RR                                    (b) FB15k-237

Figure 5: Average node-anchor distances when varying the total number of anchors $A$ from 25 to 1000 and $k$ nearest anchors per node from 5 to 50. Note that on a sparser WN18RR the gap between min and max values is much wider than of denser FB15k-237. Signs of saturation suggest that further increasing $A$ is not beneficial.

## D  EMBEDDING VISUALIZATIONS

To further study learned representations of anchors and capabilities of the encoder, we build tSNE and UMAP projections from subsamples of FB15k-237 and WN18RR based on trained models from the transductive link prediction experiments (hyperparameters listed in Table A).

For FB15k-237, we randomly sample 1000 entities (out of total 15K) and find their top-100 most common anchors. The anchor embeddings are extracted from the learned tensor while 1000 entity embeddings are obtained through the NodePiece encoder. Similarly for WN18RR, we sample 4000 entities (out of total 40K) keeping their top-100 most common anchors. As we use the RotatE decoder that assumes entities and anchors are modeled in a complex space, we visualize their real parts (e.g., first 100 dimensions out of 200).

Recalling that link prediction performance of NodePiece + RotatE retains 80-90% of the state of the art models performance, the results on Fig. 6 and Fig. 7 demonstrate that (1) NodePiece encoder is able to reconstruct clusters of similar entities; (2) anchors are well-scattered among communities. Albeit entity embeddings are built as a composition of $k$ anchors, it can be seen that all communities have "specialized" nearby anchors. On a higher level, common anchors tend to be well-scattered in the space. Less common anchors, as seen on FB15k-237 and Fig. 6, tend to group together. However, thanks to the non-linear nature of the NodePiece encoder, resulting entity embeddings still form different clusters and communities not concentrated around one point. We believe this is the effect of a compositional encoder and plan to investigate this phenomenon further.

## E  TRANSDUCTIVE LINK PREDICTION RESULTS: MRR

In addition to Figure 2 that presents Hits@10, we report variations of mean reciprocal rank (MRR) depending on combinations of $A$ and $k$ on Figure 8 from the same set of experiments. On sparser WN18RR, smaller $A/k$ combinations like 25/5 or 50/10 struggle with more precise predictions like

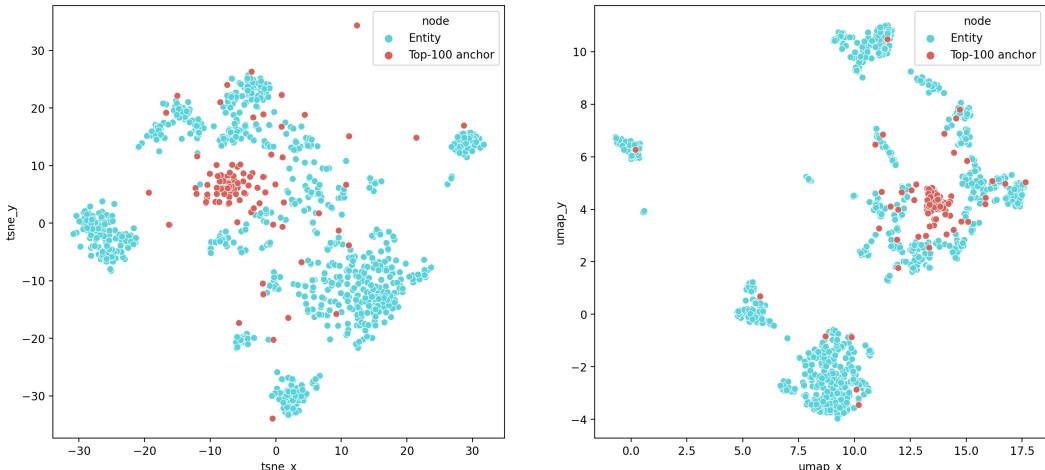

Figure 6: tSNE (left) and UMAP (right) projections of 1000 encoded entities sampled randomly from FB15k-237 and their top 100 most common anchors.

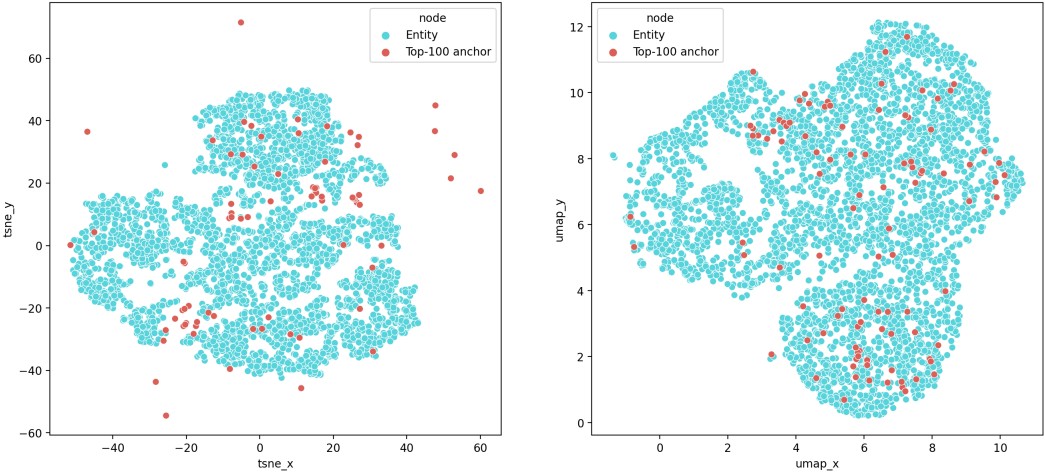

Figure 7: tSNE (left) and UMAP (right) projections of 4000 encoded entities sampled randomly from WN18RR and their top 100 most common anchors.

Hits@1 which is captured by low values of MRR. Starting from 500/10, the WN18RR performance starts to saturate. On the other hand, on denser FB15k-237, the difference between minimum and maximum MRR is less than 4 points, and performance exhibits signs of saturation already at 50/10.

# F  DATASETS CONSTRUCTION

## F.1  NODE CLASSIFICATION: WD50K NC

The original WD50K (Galkin et al., 2020) contains a triple-only KG version on which we base a new dataset for semi-supervised multi-class multi-label node classification. First, we remove all triples containing Wikidata properties P31 (*instance of*) and P279 (*subclass of*) as they already contain class information. We then remove nodes that became disconnected after removing those edges. Third, using SPARQL queries, for each remaining node in a graph, we extract a 3-hop class hierarchy of Wikidata classes and their superclasses. We only keep class labels that occur at least 50 times in the training set. Then, we sample 10% of nodes with labels for validation and 10% for test, and of remaining 80% we sample a set of nodes for the semi-supervised setup, i.e., we keep only 5% and

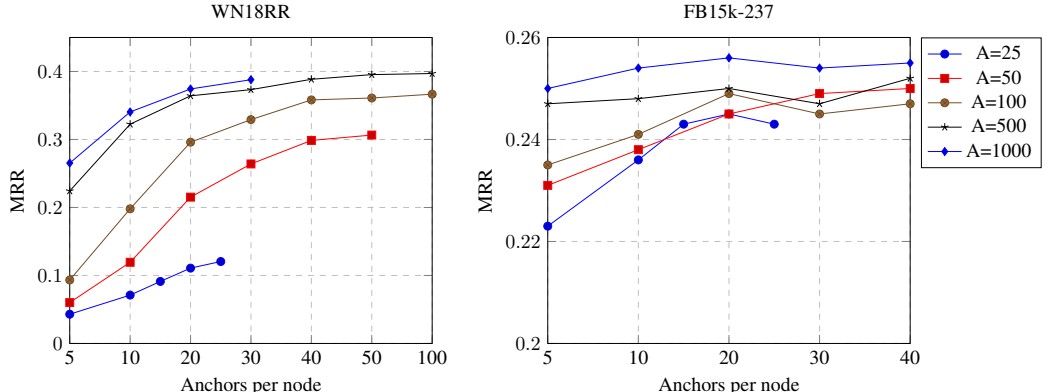

Figure 8: Combinations of total anchors $A$ and anchors per node. Denser FB15k-237 saturates faster on smaller $A$ while sparse WN18RR saturates at around 500 anchors. MRR metric captures all ranks. Note that performance gap on FB15k-237 is very small indicating that saturation has occurred already with small anchor configurations.

10% of those nodes. The resulting graph has 46k nodes, 526 distinct relation types, and 465 class labels.

### F.2    OUF-OF-SAMPLE LINK PREDICTION: OYAGO 3-10

For sampling the out-of-sample version of a bigger YAGO 3-10 we largely follow the same original procedure described in Section 4 of (Albooyeh et al., 2020). We first merge the train, validation and test triples from the original dataset for transductive link prediction. Then, from all entities appearing in at least two triples, we randomly sample 5% of nodes to be the out-of-sample entities for validation and 5% for test. All triples containing the out-of-sample entities on subject or object positions are put into validation or test, respectively, as edges that connect an unseen entity with the seen graph.

## G    NODE CLASSIFICATION: TRAINING CURVES

Figure 9 depicts train and validation values of Hard Accuracy and PRC-AUC metrics for all the compared models on WD50K NC with 5% of labeled nodes. The NodePiece model has only 50 total anchors with 10 nearest anchors per node, and 5 unique relation types in the relational context. The performance on the dataset with 10% of nodes is almost the same, so we report the charts only on 5% dataset. By the generalization gap we understand the delta between training and validation values.

The MLP baseline quickly overfits but fails to generalize on the validation. The generalization gap of CompGCN is smaller compared to MLP but is still significant, i.e., validation performance is 2–3× smaller than train. Finally, the NodePiece-enabled model has the smallest generalization gaps, especially along the Hard Accuracy metric where the validation performance is very close to that of train. Similarly, the gap on PRC-AUC is smaller than 10 points.

As shown in the ablation study in Table 6, it appears that explicit node embeddings do not contribute to the classification performance. Hence, the baseline models tend to be overparameterized where learnable node embeddings add noise, while the NodePiece model has only a few anchors (or no anchors at all when using only the relational context), much fewer parameters, and therefore generalizes better. This hypothesis also explains the observation that the node classification performance does not improve when increasing $A/k$ anchor configurations.

## H    PROOFS

**Proposition 2.** *The nearest-anchor encoder with $\binom{|A|}{k}$ anchors and $|m|$ subsampled relations, can be considered a $\pi$-SGD approximation of $(k + |m|)$-ary Janossy pooling with a canonical ordering induced by the anchor distances.*

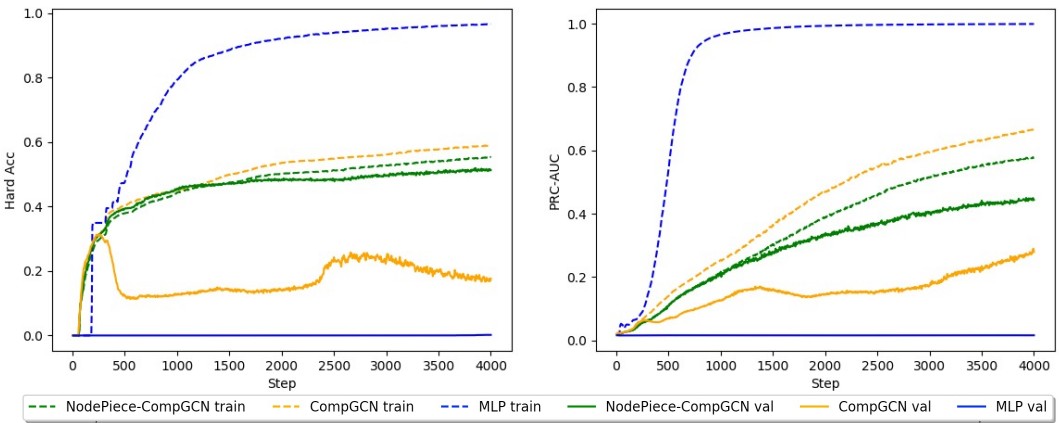

Figure 9: Generalization gap on WD50K (5% labeled nodes). NodePiece-based model has observably smaller generalization gaps compared to the baselines.

**Proof**.

We begin by providing the definition of Janossy pooling as it was presented in the original paper Murphy et al. (2019).

**Definition 1** (Janossy pooling). *Let $\mathbb{H}^{\cup}$ be the union of all anchors and relations. Consider a function $\vec{f} : \mathbb{N} \times \mathbb{H}^{\cup} \times \mathbb{R}^d \to \mathbb{F}$ on variable-length but finite sequences $\mathbf{h}$, parameterized by $\boldsymbol{\theta}^{(f)} \in \mathbb{R}^d$, $d > 0$. A permutation-invariant function $\overline{\overline{f}} : \mathbb{N} \times \mathbb{H}^{\cup} \times \mathbb{R}^d \to \mathbb{F}$ is the Janossy function associated with $\vec{f}$ if*

$$\overline{\overline{f}}(|\mathbf{h}|, \mathbf{h}; \boldsymbol{\theta}^{(f)}) = \frac{1}{|\mathbf{h}|!} \sum_{\pi \in \Pi_{|\mathbf{h}|}} \vec{f}(|\mathbf{h}|, \mathbf{h}_{\pi}; \boldsymbol{\theta}^{(f)}), \tag{3}$$

*where $\Pi_{|\mathbf{h}|}$ is the set of all permutations of the integers $1$ to $|\mathbf{h}|$, and $\mathbf{h}_{\pi}$ represents a particular reordering of the elements of sequence $\mathbf{h}$ according to $\pi \in \Pi_{|\mathbf{h}|}$. We refer the operation used to construct $\overline{\overline{f}}$ from $\vec{f}$ as Janossy pooling.*

While Janossy pooling provides a simple approach to construct permutation-invariant functions from arbitrary permutation sensitive functions, it is computationally intractable due to the need to sum over all computations. Three general strategies proposed under this framework to overcome this combinatorial challenge: canonical orderings, $k$-ary Janossy pooling, and $\pi$-SGD approximations.

A very effective way of reducing the complexity is to constrain the permutations to a canonical ordering that is independent of a specific adjacency matrix ordering over a given graph. More precisely, one defines as a function $\text{CANONICAL} : \mathbb{H}^{\cup} \to \mathbb{H}^{\cup}$ such that $\text{CANONICAL}(\mathbf{h}) = \text{CANONICAL}(\mathbf{h}_{\pi}) \forall \pi \in \Pi_{|\mathbf{h}|}$ and only considers functions $\vec{f}$ based on the composition $\vec{f} = \text{CANONICAL} \circ \vec{f}'$ (Murphy et al., 2019). In the case of NodePiece we are able to define this ordering for the anchors according to their distance to the target node. Assuming that the number of relations is fixed or grows at slow rate throughout the life-cycle of a graph we can define an arbitrary ordering for relations as a canonical ordering for the relational context. However, since anchors can be equidistant such a canonical ordering does fully satisfy permutation invariance. We propose a trivial relaxation of the original definition of canonical orderings simply requiring that an ordering greatly reduce the number of unique permutations since in practice an exact canonical ordering is rarely feasible. Specifically, $|\{\text{CANONICAL}(\mathbf{h}_{\pi}) \forall \pi \in \Pi_{|\mathbf{h}|}\}| \ll |\{(\mathbf{h}_{\pi}) \forall \pi \in \Pi_{|\mathbf{h}|}\}|$.

To further reduce the number of permutations we can truncate our ordered sequence $\mathbf{h}$. This is known as $k$-ary Janossy pooling pooling (Definition 2) and is implicitly performed by the NodePeice algorithm by varying the anchor per node parameter, $k$, and the size of the relational context, $|m|$.

**Definition 2** ($k$-ary Janossy pooling). *Fix $k \in \mathbb{N}$. For any sequence $\mathbf{h}$, define $\downarrow_k (\mathbf{h})$ as its projection to a length $k$ sequence; in particular, if $|\mathbf{h}| \geq k$, we keep the first $k$ elements. Then, a $k$-ary*

Table 15: Relation prediction results. $|V|$ denotes vocabulary size (anchors + relations).

| | FB15k-237 | | | WN18RR | | | YAGO 3-10 | | |
| --- | --- | --- | --- | --- | --- | --- | --- | --- | --- |
| | $|V|$ | MRR | H@10 | $|V|$ | MRR | H@10 | $|V|$ | MRR | H@10 |
| RotatE | 15k + 0.5k | 0.905 | 0.979 | 40k + 22 | 0.774 | 0.897 | 123k + 74 | 0.909 | 0.992 |
| NodePiece + RotatE | 1k + 0.5k | 0.874 | 0.971 | 500 + 22 | 0.761 | 0.985 | 10k + 74 | 0.951 | 0.997 |
| - no rel. context | 1k + 0.5k | 0.876 | 0.968 | 500 + 22 | 0.541 | 0.958 | 10k + 74 | 0.898 | 0.993 |
| - no distances | 1k + 0.5k | 0.877 | 0.970 | 500 + 22 | 0.746 | 0.975 | 10k + 74 | 0.943 | 0.997 |
| - no anchors, rels only | 0 + 0.5k | 0.873 | 0.971 | 0 + 22 | 0.545 | 0.947 | 0 + 74 | 0.951 | 0.998 |

*permutation-invariant Janossy function* $\overline{\overline{f}}$ *is given by*

$$\overline{\overline{f}}(|\mathbf{h}|, \mathbf{h}; \boldsymbol{\theta}^{(f)}) = \frac{1}{|\mathbf{h}|!} \sum_{\pi \in \Pi_{|\mathbf{h}|}} \vec{f}(|\mathbf{h}|, \downarrow_k(\mathbf{h}_\pi); \boldsymbol{\theta}^{(f)}). \tag{4}$$

Since an imperfect truncated canonical ordering may still result in a potentially intractable number of permutations, we use permutation sampling also known as $\pi$-SGD to learn arbitrary functions that approximate $(k + |m|)$-ary Janossy pooling. This is done by randomly ordering anchors that are equidistant resulting in a uniform sampling of possible permutations during training and evaluation. For more details on the formal definition of $\pi$-SGD we point the reader to the original paper (Murphy et al., 2019).

# I    RELATION PREDICTION

**Setup.** We conduct the relation prediction experiment on the same FB15k-237, WN18RR, and YAGO 3-10 datasets. While link prediction deals with entities, the relation prediction model has to rank a correct relation given a *(head, ?, tail)* query. We report MRR and Hits@10 in the filtered setting as evaluation metrics. Similar to the link prediction configuration, we use NodePiece + 2-layer MLP and compare against RotatE of the same total parameter count.

**Discussion.** The reported results (Table 15) demonstrate a competitive performance of NodePiece-based models with reduced vocabulary sizes bringing more than 97% Hits@10 across graphs of different sizes. In the case of WN18RR and YAGO 3-10, NodePiece models with fewer anchors even slightly improve the accuracy upon the shallow embedding baseline. The ablation study suggests that on dense graphs with a reasonable amount of unique relations having explicit learnable node embeddings might not be needed at all for this task. That is, we see that on FB15k-237 and YAGO 3-10 the NodePiece hashes comprised only of the relational context deliver the same performance without any performance drop confirming the findings from the previous experiment.

# J    INDUCTIVE LINK PREDICTION

**Setup**. Nodes in the inference graphs do not have any associated feature vectors which makes this benchmark very relevant for graph representation learning. Importantly, the set of relation types in the inference graphs is a subset of those seen in the training set. Since the relation embedding matrix can be learned on the training graph, we therefore have a uniform method for constructing node representations both on seen and unseen graphs. As an encoder we try MLP and Transformer.

**Evaluation Protocol**. Following the original work (Teru et al., 2020), we employ a filtered setting and rank each triple against 50 random negative triples reporting the Hits@10 metric. This setup is motivated by computational complexity of GraIL at inference time, while NodePiece + CompGCN is as fast in the inductive inference as in the transductive regime reported in other experiments.

**Discussion**. For each KG, there are 4 splits of increasing size of train and inference nodes and edges. The empirical results on this spectrum of various sizes demonstrates interesting scalability properties of NodePiece in inductive settings. Without anchor nodes, the NodePiece vocabulary size is independent of the number of nodes and edges, depending only on the number of relation types. On dense relation-rich graphs such a vocabulary is enough to yield very competitive performance.

Table 16: Hyperparameters for inductive link prediction experiments. Entries are shared among 4 splits of each graph if not particularly specified. V1 | V2 | V3 | V4 otherwise.

| Parameter | FB15k-237 | WN18RR | NELL-995 |
|---|---|---|---|
| # Anchors, $|A|$ | - | - | - |
| # Anchors per node, $k$ | - | - | - |
| Relational context, $m$ | 12 | 4 | 4 | 4 | 3 | 4 | 6 | 4 | 6 |
| Vocabulary dim, $d$ | 100 | 100 | 100 |
| Batch size | 512 | 512 | 512 |
| Learning rate | 0.0001 | 0.0001 | 0.0001 |
| Num negatives | 32 | 32 | 32 |
| Epochs | 2500 V1 | 2000 rest | 590 | 2000 | 210 | 2000 | 2000 |
| NodePiece encoder | MLP | MLP | MLP | MLP | MLP | Trf |
| NodePiece encoder dim | 200 | 200 | 200 |
| NodePiece encoder layers | 2 | 2 | 2 |
| NodePiece encoder dropout | 0.1 | 0.1 | 0.1 |
| CompGCN layers | 3 | 3 | 6 | 6 | 10 | 3 | 4 | 3 | 3 |
| CompGCN attention | yes | yes | yes |
| CompGCN dropout | 0.1 | 0.1 | 0.2 | 0.1 | 0.1 | 0.1 |
| Loss function | NSSAL | NSSAL | NSSAL |
| Margin | 25 | 15 | 15 | 25 | 15 | 15 | 5 | 20 | 15 | 20 | 30 | 20 |
| Training time, hours | 2 | 4 | 16 | 19 | 1 | 18 | 4 | 10 | 6 | 5 | 8 | 8 |

## K  UNIQUENESS OF NODE HASHES

NodePiece represents nodes as a sequence of tokens and a natural question in this context is how unique such sequences can be in light of different anchor selection and tokenization strategies.

Assuming the input graph is a single connected component, when sampling $|A|$ total anchors and selecting *randomly* $k$ anchors per node, the number of possible hash combinations is bounded by $\binom{|A|}{k}$. In this scenario, uniqueness of hashes is achieved by having this number bigger than the number $N$ of nodes in a graph, $\binom{|A|}{k} > N$, and this happens with high probability for any reasonably large $|A|$, eg, $\binom{50}{20}$ encodes about $4.7 \cdot 10^{13}$ combinations which covers all existing public KGs combined.

In the deterministic selection of nearest anchors per node we do not have such guarantees. Nevertheless, additional sequences of relational contexts and anchor distances help to obtain more unique hashes. Collisions are possible in highly regular graphs like Wordnet, but real-world KGs like Wikidata and DBpedia do not exhibit such a regular structure. Similar to homonyms whose meaning depends on the surrounding context in a sentence, we hypothesize that adding message passing layers (that can be seen as encoding of a neighboring context for a given node) on top of NodePiece hashes might further improve the diversity of node representations. That said, the future work research agenda might include proving tighter theoretical bounds on hashes uniqueness and developing new anchor sampling and tokenization strategies.

