# OpenReview forum: "NodePiece: Compositional and Parameter-Efficient Representations of Large Knowledge Graphs"
_ICLR.cc/2022/Conference — ICLR 2022 Poster_

### Official Review · Reviewer_X7aq · 2021-10-26

**Correctness:** 4
**Technical Novelty And Significance:** 3
**Empirical Novelty And Significance:** 3
**Recommendation:** 6
**Confidence:** 4

**Main Review:**

### Strength
Finding new deep-learning architectures and strategies that are more efficient and usable with lower computational resources without sacrificing performance is a noble goal, and it can improve the actual applicability of neural knowledge-graph based methods. The paper considers a number of trade-offs and heuristics for constructing compositional-vertex embeddings and presents either SOTA or close to the SOTA results on a number of benchmarks. I believe that this paper will be much appreciated by practicioners.

### Weakness
1. The paper repeatedly claims the uniqueness of the hashes: based on my reading of the paper none of the hashing strategies considered in the paper can guarantee that two distinct nodes will always have distinct hashes. However the paper claims that the hashes are unique in multiple paragraphs.
2. Claim of sub-linear vocabulary size: This may be a pedantic objection, but based on my reading of the results in figure 2, and table 4 , the model requires roughly 10% of the nodes. The experiment in section 4.1.1 does suggest that using only 1% of the nodes was sufficient for encoding the 2.5M subset of wikipedia, but I would have preferred that the authors use a larger set of 25M nodes or all 100M of them before making claims of sub-linear growth in parameters. It is useful to maintain the distinction between linear with a small constant factor and sub-linear growth and I don't see compelling evidence for the latter.


**Summary Of The Paper:**

Conventional methods for learning knowledge graph embeddings often learn separate embeddings for each vertex in a knowledge graph. This paper presents empirical results about a new heuristic for encoding the vertices in a knowledge graph which drastically reduces the number of parameters in a graph, and allows better handling of novel vertices, whose connections were not known at the time of learning/training.

  The basic approach is to encode each vertex through a sketch of its neighborhood. Specifically the paper heuristically selects a subset of training vertices, called "anchors". Each vertex is represented by its closest k-anchor vertices, and the types of the outgoing edges starting from the vertex.

  Individual embedding vectors are learnt only for the anchor vertices, and at inference time, the embedding of a novel vertex is composed from the closest k-anchor vertex embeddings using some neural network layer such as an MLP or a transformer.


**Summary Of The Review:**

The paper presents a novel approach for embedding large knowledge graphs using a small number of parameters. The empirical results strongly support the usefulness of the proposed approach and I believe that even though this won't be the final word on creating compositional representations of KG vertices, but it will be appreciated by ICLR readers.

---

> ### Author Response · Authors · 2021-11-12
> **Response to Reviewer X7aq**
>
> We thank the Reviewer for the inspiring feedback as to the novelty of the approach and compelling empirical results. Indeed, with NodePiece we hope to lower the computational burden when training on large KGs and allow practitioners without expensive 32/48 GB GPUs to fit the model parameters onto consumer GPUs. Perhaps more importantly, NodePiece enables on the fly generation of node embeddings after training allowing for use in industrial settings where KGs are dynamic (new nodes connecting to the main graph might appear over time) and latency requirements do not permit the retraining of the entire embedding matrix (e.g. recommending personalized movies to a new user within minutes rather than hours).
>
> We would like to address the weak points:
> 1. > The paper repeatedly claims the uniqueness of the hashes
>
> We understand the concern on the uniqueness of hashes and should have clarified that for different tokenization strategies. Generally speaking, assuming the graph is a single connected component, when sampling $|A|$ total anchors and selecting _randomly_ $k$ anchors per node we would have ${|A|\choose k}$ possible combinations of hashes. In this scenario, we just want to make sure that this number is bigger than the number N of nodes in a graph, ${|A|\choose k} > N$ , and this happens with high probability for any reasonably large |A|, eg, 50 choose 20 encodes 4.7 * 10^13 combinations which covers pretty much all existing public KGs combined.
>
> In the deterministic selection of nearest anchors per node we do not have such guarantees and that’s where additional sequences of relational contexts and anchor distances help to obtain more unique hash sequences. Indeed, collisions are possible in highly regular graphs but real-world KGs are not that regular (among all graphs in the experiments maybe WordNet WN18RR is the most regular tree-like). Generally, the problem of unique node identification is a big research area involving Weisfeiler-Lehman tests and various ways to get positional encodings for GNNs. We’d definitely like to apply those mechanisms to rigorously quantify the uniqueness of hashes in the deterministic selection scenario in future work, thanks for the suggestion!
>
> 2. > Claim of sub-linear vocabulary size
>
> We tried to be careful with sublinearity and only mentioned it once in the abstract in the hypothesizing context “possibly, would be great to have”. Although we do not have a solid proof for sublinear scaling as we admit in Section B of the Appendix, our experiments indicate that a constant-size vocabulary of only relation types can perform well on node classification and relation prediction. Please have a look at the new result on inductive link prediction (in the general response) where the inference graph is a new graph of new nodes disconnected from the train graph and we predict links among unseen nodes, such that we can’t benefit from anchor nodes and only learn a vocabulary of relation types. This yields good results with a constant size vocabulary on dense relation-rich graphs like Freebase and NELL. And for each graph we evaluate NodePiece on 4 different splits of different training/inference graph sizes.
>
> Having said that, in the revised version we removed the mention of sublinearity from the abstract to avoid further confusion and only elaborate on that in Section B of the Appendix.

---

> ### Author Response · Authors · 2021-12-03
> **Discussion**
>
> Dear Reviewer X7aq, please let us know if our response addressed your concerns or if you have any further questions.

---

### Official Review · Reviewer_2qcD · 2021-11-02

**Correctness:** 3
**Technical Novelty And Significance:** 3
**Empirical Novelty And Significance:** 3
**Recommendation:** 8
**Confidence:** 4

**Main Review:**

More specifically the proposed method (NodePiece)  first (randomly) selects a fraction of all entities as anchors, which have learnable embeddings. Then the representation of a target entity consists of a concatenation of its nearest (sampled) anchors' embeddings (shifted by distance embeddings) and embeddings of (sampled) relations which this entity possesses. The concatenated representation is transformed by either a MLP (for model quality) or Transformer (for model compactness).

For link prediction, MRR and H@10 for NodePiece is much worse than RotatE on small and bigger KGs. On a large scale KG (OGB WikiKG 2) of 2.5M nodes NodePiece improves a full embedding based baseline (AutoSF) in both model size and MRR.

For node classification, NodePiece is competitive to an existing approach (RotatE) with more compact models.

For relation prediction, NodePiece is significantly better than previous approach (CompGCN) on WD50K with either 5% or 10% labeled data. However, under the out-of-sample link prediction setting, NodePiece is significantly worse than the previous approach oDistMult-ERAvg on oFB15k-237.   The comparison on oYAGO 3-10 (117k) was not achieved due to memory issues.


Overall this work represents a promising direction with some promising results. However, the proposed approach seems to be flawed and shows degraded quality in some of the situations.  It seems that the proposed randomized anchor selection can lead to suboptimal representation. Since some of the key nodes (e.g., entity types) might be missed. It is probably better to have a learning based approach for anchor selection.

Minor issues:

Eq 1 and Eq 2: why are they called "hashes"? these representations are just learnable embeddings right?

Is it a typo for "injective encoder function enc"? I am not sure that it is possible to have injective functions  for R(k+m)×d → Rd.

Detail is needed on how the proposed node representations are used for link predictions. Do we concatenate the presentation of two nodes?

Detail is needed on how  the link prediction metrics are calculated.
What negative triples are we ranking against? How are the true triples corrupted?



**Summary Of The Paper:**

Previous KG representation learning approaches learn an embedding for each entity in KG, which leads to a lot of parameters in the models. This work aims to avoid this problem by dynamically generating the embedding of an entity based on its local neighborhood.

**Summary Of The Review:**

A promising direction with mixed results.

After author response: I am satisfied with the answers to all my questions.

---

> ### Author Response · Authors · 2021-11-12
> **Response to Reviewer 2qcD**
>
> We thank the Reviewer for the comments and acknowledging a new promising direction laid by NodePiece for dynamically generating embeddings based on their local neighborhoods. We would like to address the raised concerns.
>
>  > the proposed approach seems to be flawed and shows degraded quality in some of the situations
>
> In several experiments, NodePiece-enabled models indeed fall behind state-of-the-art by some margin. Still, the major idea of this work is about a uniform way to compose node embeddings applicable in both transductive and inductive cases - exactly what the compared baselines cannot do as all of them are inherently transductive since they learn node embeddings (thus cannot transfer onto graphs of unseen nodes).
> We’d outline that some performance loss is inevitable in this trade-off but we believe that the amount of positive benefits brought by the NodePiece compositional representation learning strategy might outweigh those losses as we show that NodePiece features perform reasonably well on a wide variety of common KG tasks.
>
> To further support this claim, we added results of a new experiment on inductive link prediction (in the paper and in the general response) where the graph at inference time consists of new, unseen entities and prediction has to happen among unseen entities. In addition to reporting the numbers in the experiments we identify several interesting points: (1) NodePiece features reach reasonable and competitive performance at a fraction of parameter budget of shallow models; (2) relational context is an underestimated concept that can bring significant benefits to many practical tasks.
>
> > proposed randomized anchor selection can lead to suboptimal representation. Since some the key nodes (e.g., entity types) might be missed
>
> Please note that in all the experiments anchors are selected according to the mostly deterministic strategy (40% top degrees, 40% top pagerank, 20% random), and we elaborate on this decision in Sections C and D of the appendix. With top degree and PR centrality measures the key nodes (along those measures) are likely to be inserted in the anchor set.
>
> As to the entity types, you actually raise an important question. That is, in standard node classification tasks (like in OGB) we assume that node labels are not actual nodes in a graph. In KGs, however, graph schema might reside in the same space as factual data, eg., a triple <USA, isA, Country> connects a node USA to a class Country via a specific relation. Class nodes quickly become outliers from the graph topology point of view - they might have thousands or millions of incoming edges but only few outgoing (if there is any class hierarchy). Following the common practice in the GNN community, we sampled WD50k datasets from Wikidata by removing entity type nodes and their respective relation types to prevent any leakages. We elaborate on the sampling strategy in Section F in the appendix.
>
> > It is probably better to have a learning based approach for anchor selection
>
> This is a great suggestion and we definitely want to explore this area in the future work. In this paper, we show that using anchors and relational context generally yields reasonable results in many tasks with the benefit of generalization to unseen nodes and low compute cost. The next step surely includes finding more efficient anchor selection and feature composition strategies. This is a non-trivial objective as it includes parameterizing discrete sampling before we know any downstream task. For years in NLP, subword token vocabularies were built in a deterministic frequency-based way, and only very recently some learning based methods have been proposed [1].
>
> > Eq 1 and Eq 2: why are they called "hashes"? these representations are just learnable embeddings right?
>
> We refer to the term “hashing” to illustrate the fact that nodes are first tokenized in a sequence of discrete IDs, namely, anchors and relations. For instance, if we have total 10 anchors and 5 relation types, a node hash of 5 anchors and 3 relations might look as [3,9,2,1,4,  0,3,1]. Then, those IDs are passed through anchor and relation embedding layers, get vectorized, and encoded into one resulting vector.
>
> > Is it a typo for "injective encoder function enc"? I am not sure that it is possible to have injective functions for R(k+m)×d → Rd
>
> Thanks for highlighting this point. Injectivity depends on the encoder. Surely, the MLP encoder that projects a concatenated $(k+m)d$ vector to $d$ is not injective, but for set poolers like DeepSets or Transformer there exist certain theoretical guarantees (cf Lemma 4.2 in [6]) that an encoder is injective as long as its dimension is higher than the number of input elements. In our practical cases, hash length is always less than embedding dimension, so we can assume it will be injective. Still, to avoid further confusion, we removed the word “injective” from this sentence in the revised version.

---

> > ### Author Response · Authors · 2021-11-12
> > **Response to Reviewer 2qcD Part II**
> >
> >
> > > Detail is needed on how the proposed node representations are used for link predictions. Do we concatenate the presentation of two nodes? Detail is needed on how the link prediction metrics are calculated. What negative triples are we ranking against? How are the true triples corrupted?
> >
> > For link prediction (and relation prediction, Table 6) experiments we follow a standard training protocol in the KG embedding domain defined in the classical work of Bordes et al [2]. Given a triple [head, relation, tail] we first obtain head and tail representations via NodePiece, extract an embedding for the relation, and apply a non-parametric scoring function - RotatE [3] in our case - to those three vectors to get a score for the triple. In the training regime with negative sampling, each batch of true triples is accompanied by a batch of negative triples where either a head or tail of each triple is randomly replaced by a random entity ID existing in a graph. We sample N negative triples per each true triple where N/2 triples would have corrupted heads and N/2 would have corrupted tails. The negative triples are passed through the same non-parametric scoring function as well.
> >
> > The training objective is to ensure that scores for true triples are higher than for negative triples (this can be done, e.g., with a margin loss or binary classification loss). At the evaluation step, for each testing triple [h,r,t] we first compute scores of all entities in the tail position [h, r, ?] and find the score rank of the true tail node t. We then repeat the procedure for all entities in the head position [?, r, t] looking for the score of the true head h of the input triple. Note that the we use a common filtered setting defined in [2], that is, if a triple [h,r,t] has other true answers [h,r,t1] or [h,r,t2] we will mask the scores of t1 and t2 setting them to -inf such that they won’t interfere in the ranking process. Please refer to the recent surveys [4, 5] for more training details - we are glad to elaborate if you have any further questions.
> >
> > [1] Xu et al. Vocabulary Learning via Optimal Transport for Neural Machine Translation. ACL’21
> > [2] Bordes et al. Translating Embeddings for Modeling Multi-relational Data. NIPS’13.
> > [3] Sun et al. RotatE: Knowledge Graph Embedding by Relational Rotation in Complex Space. ICLR’19
> > [4] Ruffinelli et al. You CAN Teach an Old Dog New Tricks! On Training Knowledge Graph Embedding, ICLR 2020
> > [5] Ali et al. Bringing light into the dark: A large-scale evaluation of knowledge graph embedding models under a unified framework. TPAMI 2021
> > [6] Wagstaff et al. On the Limitations of Representing Functions on Sets. ICML’19

---

> ### Comment · Reviewer_2qcD · 2021-11-25
> **Thanks for the answers.**
>
> I am satisfied with the answers to all my questions.

---

### Official Review · Reviewer_CpaB · 2021-11-02

**Correctness:** 4
**Technical Novelty And Significance:** 2
**Empirical Novelty And Significance:** Not applicable
**Recommendation:** 5
**Confidence:** 4

**Main Review:**

Strength:

1. Experiment on a large-scale link prediction dataset (Table 5) shows encouraging results on the effectiveness of NodePiece.

Weakness:
1) The paper has only focused on graphs with multi-type relations (knowledge graphs). If the main idea of NodePiece is to compress embeddings, why not test it on graphs with only one type of relationship (e.g., large citation networks)? It would be great to see if such heuristic apply to single-type datasets and whether it could help methods such as transductive unsupervised embedding learners (e.g., DeepWalk).

2) When NodePiece shows improvement over baselines, i.e., node classification and relation classification (Tables 6-7), relational context alone does well too. The fact that anchor nodes are not useful in these tasks is probably an indicator of the triviality of the tasks. – Based on this, I doubt if NodePiece will do well on single-type relation graphs (first weakness).

3) The paper emphasizes that the goal of NodePiece is not to improve performance of the baselines but rather highlight efficiency in memory and maintaining a reasonable accuracy. However, I think the paper doesn’t do justice in explaining the experimental observations. For instance, it is not clear why increasing the anchor nodes does not improve the performance beyond a point (saturation happens). That is, by increasing the number of parameters and memory, shouldn’t the performance also get better? Analogously, what is special about WikiKG dataset that leads to such good result?

Question:
1. What is the reason to create inverse edges? It is mentioned that it is due to maintaining reachability and the balancing of in and out degrees. Could you please elaborate?

2. Did any of the experiments include node features?



**Summary Of The Paper:**

This paper presents NodePiece, a method to scale up GNNs by means of removing their dependence on individual node embeddings which grow linearly in size and are inefficient in huge graphs. The method uses a set of anchor nodes, which are picked from the graph itself, for the NodePiece representations. Concretely, NodePiece embedding is derived by the embeddings of the closest anchor nodes and the hop-based distance to these nodes which are also represented by vectors. – the maximum distance between two nodes in a graph is Diameter(G), so the number of distance/position embeddings is Diameter(G). Additionally, NodePiece is augmented by a sample of embeddings of the relationship types the node is involved in. Apart from induced efficiency, NodePiece can be especially useful in inductive learning where an unseen node’s embedding can be created through the anchor nodes.

**Summary Of The Review:**

The paper is inspired by the use of subwords in NLP and aims to leverage similar techniques for GNNs. Evidently, the proposed method is not behaving like subwords, as NodePiece could potentially degrade the performance a lot. There are missing experiments, and the paper has weaknesses in its analysis of the performance of NodePiece, i.e., we don't know when it works and when it does not.

---

> ### Author Response · Authors · 2021-11-12
> **Response to Reviewer CpaB**
>
> We thank the Reviewer for the valuable feedback and acknowledging the applicability of NodePiece in inductive learning cases - that was indeed one of the main motivations behind our approach.  Several questions appeared due to some unclearly explained parts in the paper - we admit that and would like to clarify these points in this response.
>
> 1. > why not test it on graphs with only one type of relationship
>
> NodePiece is designed for multi-relational graphs (eg, KGs), their specific challenges, and is evaluated on various KG representation learning tasks. The main reason why we focus on KGs is that such graphs often do not have node features (especially those not grounded in language) and pretty much all existing pipelines for KG tasks imply learning shallow embedding vectors. This is a big computational burden and is in contrast to standard GNNs tasks (like those in OGB node classification for instance) where nodes already have rich features and we mostly learn GNN weights to properly deal with those features. Rich node features also enable easy transfer to inductive learning tasks common in the GNN literature with new graph structures (but features are known in advance as input).
>
> For KGs, however, inductive learning in the absence of node features is non-trivial. This challenge demands certain trade-offs. In our case, NodePiece offers a uniform mechanism for composing representations for transductive and inductive tasks (link prediction, node classification, relation prediction) for a price of somewhat lower transductive LP performance.
>
> It is generally possible to apply NodePiece for single-relational networks but it doesn’t address specific challenges of such networks. Current unsupervised approaches like InstantEmbedding [1] or FREDE [2] greatly improve over DeepWalk & node2vec by allowing for local node embedding reconstruction in sublinear time for billion-scale graphs. They are great tools for addressing scalability challenges of single-relational networks but can’t be easily adapted for multi-relational graphs. Although NodePiece is not backed in matrix factorization theory, conceptually it aims at a similar objective for KGs, allowing for new embeddings to be generated on the fly. That’s why our experimental agenda targets KG tasks.
>
> As touched on earlier, NodePiece is not designed for scaling GNNs -- the main challenge in that domain is sampling rich neighborhoods without exponential size explosion while retaining message passing expressiveness. Node features in those tasks are generally given beforehand, too. A good example of solving such a challenge is GAS [3]. In KGs we often do not have node features, so the first problem (before tackling any neighborhood sampling) is representation learning of good features, and that is where NodePiece can help. In fact, NodePiece and GAS are orthogonal enough to be paired together in cases where we do not have pre-computed node features and want expressive scalable message passing GNN encoders on top of NodePiece features.

---

> > ### Author Response · Authors · 2021-11-12
> > **Response to Reviewer CpaB Part II**
> >
> >
> > 2. > The fact that anchor nodes are not useful in these tasks is probably an indicator of the triviality of the tasks
> >
> > We understand that looking at Table 6 (Relation Prediction, RP) it might seem as an already saturated benchmark. With the same training protocol as link prediction, in RP evaluation we rank all possible _relations_ given a query (h, ?, t). Surely, the number of distinct relations in KGs is much smaller than the number of entities, i.e., 474 (incl. inverses) for FB15k-237 vs 15k nodes, or just 22 relations in WN18RR vs 40k nodes, which conversely leads to easier ranking and higher numbers. The point of this experiment was to show that relation-based features in relation-rich KGs (ie, Wordnet with 11 relations is not relation-rich)  are strong performers and should be treated as first-class citizens, too.
> >
> > Having said that, please check the general response where we report NodePiece performance on a newly added inductive link prediction benchmark where the graph arriving at inference time is a new graph composed of entities unseen during training. Using only relational context features we significantly improve over the recently published baselines in relation-rich KGs. We plan to replace the relation prediction experiment with this new inductive LP experiment in the camera-ready version (and move RP to the appendix).
> >
> > As to the Node Classification experiment (Table 7), the setup and objective (multi-class multi-label semi-supervised classification against 465 labels) seem to be rather hard for the MLP baseline which yields random predictions indicating it is quite a non-trivial task even when learning a full node embedding matrix. Message Passing GNN tackles the task better, but CompGCN over shallow learnable node embeddings confirms the key problem: it’s not just the neural net architecture which is important but the inputs to the net themselves and how discriminative those features are. This is a representation learning problem where NodePiece helps: we show that a few neighborhood-based features composed together carry more discriminative power compared to a much larger but shallow node vocabulary. Still, this benchmark is far from being solved as PRC-AUC and Hard Accuracy (Exact Match of a 465-d labels vector) metrics are in the 45-55% range.
> >
> > > I doubt if NodePiece will do well on single-type relation graphs
> >
> > Please refer to p.1 of this response where we advocate that performance on single-relation networks is different from the original design decisions behind NodePiece and representation learning challenges it addresses.

---

> > > ### Author Response · Authors · 2021-11-12
> > > **Response to Reviewer CpaB Part III**
> > >
> > >
> > > 3. > I think the paper doesn’t do justice in explaining the experimental observations, e.g., why saturation happens
> > >
> > > Due to the 5 reported experiments and space limitations, we kept the important ablations in the main section and, unfortunately, had to move some qualitative analysis to the appendix. The ablations aim at explaining why and when NodePiece works or does not at some graphs - for instance, WordNet WN18RR is a very sparse graph with only few unique relations, and that’s why anchors + relations form better features than purely anchor-only or relation-only ones. In fact, relation-only features drop performance to zero on WN18RR, but in relation-rich graphs it’s a completely opposite picture.
> > >
> > > Discussing the saturation phenomenon (why performance doesn’t get better when increasing parameters count), we acknowledge that at the current stage it is hard to quantify the “capacity” and discriminative power of obtained features in a rigorous way. It is a very interesting avenue for theory-based future work and we certainly see how we could contribute to it by studying NodePiece features. Nevertheless, we have several empirical “proxy” evidence we’d like to put your attention to (please excuse us for using “jargon” formulations below for intuition purposes), namely, Figures 4 and 5 in Section C of the appendix.
> > >
> > > * Figure 4 shows that increasing total anchors skews the distribution of distances only up to a certain point after which the difference is less pronounced. For relation-rich and denser FB15k-237 the mean of the distribution stabilizes already at 50 total anchors (500 for WN18RR) which corresponds well to the observed saturation thresholds on those datasets.
> > > * Figure 5 illustrates that varying k nearest anchors per node affects the average anchor distance for each node in a graph. More importantly, note a different range of values on Y axis: [2.5, 6.5] for sparse Wordnet and only [1.2, 2.6] for denser Freebase. We thus hypothesize that saturation happens faster on dense graphs where we keep adding anchors in a close proximity that do not carry important information (i.e. that help distinguish a node from a nearby node with similar anchors). For sparser graphs, close proximity is not enough, and we need a more diverse set of anchors - again, up to a certain extent where all nodes would have the same distribution of anchor distances.
> > > * Finally, embedding projection visualizations on Fig 6 and 7 show that saturated configurations allow for the localization of anchors in their cluster - it’s better visible on the Freebase example where each community has at least one “representative” anchor. Adding more anchors would most likely only “nudge” composed representations of individual nodes (note a central group of anchors on the Fig 7) towards being in one or another community.
> > >
> > >
> > > > what is special about WikiKG dataset
> > >
> > > As to the general parameter count discussion, increasing parameter budget does not always lead to better performance as we show in the WikiKG 2 experiment (Table 5) where 100x larger models lag behind a model with NodePiece features. The WikiKG 2 benchmark has a slightly different evaluation procedure compared to smaller datasets. At inference time the evaluator ranks a triple against 1000 negatives pre-selected by dataset authors, not against all 2.5M nodes.
> > >
> > > Questions:
> > >
> > > > What is the reason to create inverse edges? It is mentioned that it is due to maintaining reachability and the balancing of in and out degrees. Could you please elaborate?
> > >
> > > KGs are directed graphs where some nodes might be hubs, or sinks, or have outgoing-only / incoming-only edges. Adding inverse edges is a standard practice in various KG embedding models learned on link prediction for mitigating such issues and to balance the in- and out-degree . For NodePiece it is of particular use in both anchor search and building a relational context. When finding nearest anchors, BFS expands outgoing edges. In a graph w/o inverse edges if a node only has outgoing edges it can never be reached from any anchor thus it cannot be represented as a combination of anchors. Similarly for the relational context, if a node has many more incoming relation types than outgoing, its outgoing relational context will be poor and will not reflect the relational neighborhood. In our preliminary studies we found that mixing incoming/outgoing relations in the relational context leads to inferior results, thus we sample either only incoming or outgoing relation types. Adding inverse edges (with inverse relation types) helps to construct and maintain a representative sample of relational neighborhood.
> > >
> > > > Did any of the experiments include node features?
> > >
> > > None of the 5 experiments have node features as discussed in Part 1 of this response.

---

> > > > ### Author Response · Authors · 2021-11-12
> > > > **References to the previous parts**
> > > >
> > > >
> > > > [1] Postăvaru et al. InstantEmbedding: Efficient Local Node Representations, 2020
> > > > [2] Tsitsulin et al. FREDE: Linear-Space Anytime Graph Embeddings, 2020
> > > > [3] Fey et al. GNNAutoScale: Scalable and Expressive Graph Neural Networks via Historical Embeddings, 2021

---

> ### Comment · Reviewer_CpaB · 2021-11-27
> **Thank you for the response**
>
> The response helps with situating the paper among related work. The response also answers most of my questions.
>
> NodePiece tackles a particular application of GNNs which no prior work has targeted specifically: inductive learning in knowledge graphs where node features are not available. The paper does contain experiments on other setups, but the results are not as strong or could be regarded as negative results (e.g., transductive link prediction). The response doesn't answer why NodePiece wasn't tested on graphs with single type edges and regards it out of the scope of the paper -- the authors have provided new results on a setup where anchor nodes cannot be used. Single-type graphs are the inverse of this setup where relational contexts cannot be used.

---

> > ### Author Response · Authors · 2021-11-28
> > **Comment**
> >
> > Thank you for the comments.
> >
> > NodePiece can indeed be applied successfully to both transductive and inductive settings, with GNNs and non-GNN models. NodePiece’s novelty and strength lies in its ability to bridge this binary framing and allow for node representation on evolving large-scale knowledge graphs in an efficient, simple, and compositional manner. Existing approaches require a combination of methods that cannot readily or easily be applied to real-world dynamic applications.
> >
> > Regarding the strength of the transductive link prediction results we would like to point the reviewer to our experiment on WikiKG2 where we achieve SOTA results, so the tradeoff of using NodePiece features is not so easily characterized as being either positive or negative. Furthermore, none of the baseline WikiKG2 models can ever be used in the inductive/dynamic setting.
> >
> > Regarding single edge type graphs, in both application and in theoretical works simple graphs and knowledge graph problems are generally treated as separate tasks. Most do not consider both concurrently because they seek to take advantage of the inductive biases unique to one class of problem. In this work we focus on the challenges of KG representation learning (i.e. capturing the relational semantics which are not present in simple graphs). Tackling simple graphs would require an experimental agenda on large-scale setups where already existing methods like InstantEmbedding are tailored to the specific challenges of this task (i.e., it deserves its own research paper).  We elaborated on our motivation in the Part I of our response. Please let us know if you have any questions left unanswered by our response here and in Part I regarding our position on simple graphs.

---

### Official Review · Reviewer_KuBz · 2021-11-02

**Correctness:** 4
**Technical Novelty And Significance:** 4
**Empirical Novelty And Significance:** 4
**Recommendation:** 8
**Confidence:** 3

**Main Review:**

Overall, the paper is clearly written and the NodePiece approach is well-motivated. The diverse range of experiments is a strong point of the work, as they demonstrate the usefulness of NodePiece for a range of important tasks including link prediction, relation prediction, and node classification, including in the inductive setting.

The primary shortcoming of the work is the limited number of comparisons to alternative methods in the experiments. In each case, only one or two existing methods are compared to NodePiece in terms of both performance and parameter cost, although the baselines do seem to be methods that are competitive with state-of-the-art approaches for each task. The ablation analyses performed in each setting could also have been limited to a few relevant tasks, in order to be able to include results that explore other design choices such as how sampling strategies for anchor nodes and outgoing relations affect downstream task performance (some of this analysis seems to have been moved to the appendix).

**Summary Of The Paper:**

This paper presents NodePiece, a method inspired by subword embeddings in NLP that is designed for constructing compositional representations of entities in a knowledge graph using a fixed-size entity and relation vocabulary. This allows for learning and storing entity representations with a number of parameters that does not scale with the size of the knowledge graph, as well as being able to construct representations for unseen entities. The method uses a vocabulary that combines relation embeddings for all relations in the knowledge graph with a limited set of anchor entities, typically much fewer than the total number of entities. To construct an entity representation, a subset of that entity's nearest neighbor anchor entities (along with embeddings of the lengths of the shortest paths to those entities) are combined with a sampled set of adjacent relation embeddings and passed through an encoder (either an MLP or a Transformer) to output a fixed-size embedding. These compositional entity representations can then be used in classicial knowledge graph tasks such as link prediction, relation prediction, and node classification. Across all of these tasks and a range of datasets, NodePiece representations are shown to achieve a large fraction of the performance of strong baselines and in some cases even outperform them, all while using a fraction of their parameter count. Ablation analyses show that increasing the total number of anchors and number of anchors per entity improves performance up to a saturation point, and that in some cases a vocabulary of relations without anchor entities is sufficient (or even preferable) for constructing well-performing entity representations for certain downstream tasks.

**Summary Of The Review:**

Though the experiments could include more baselines and further analysis of how design decisions for NodePiece affect task performance, the current set of results seems to sufficiently demonstrate the utility and versatility of the method.

---

> ### Author Response · Authors · 2021-11-12
> **Response to Reviewer KuBz**
>
> We thank the Reviewer for appreciating the clarity, motivation and experimental diversity of our work. We are glad to see the acknowledgement of the utility and versatility of NodePiece for a variety of KG representation learning tasks.
>
> The decision behind a smaller number of baselines has been taken in favor of tasks breadth: instead of comparing many existing models up to precise percentages (e.g., the transductive KG link prediction literature is enormous, we cited two recent papers each evaluating 20+ models) we’d rather provide a reference point of what a user could expect when applying NodePiece in a particular task. Given that the compared models are competitive with state-of-the-art, the reported numbers could serve as a good estimate for adjusting to a specific more/less expressive model.
> Nevertheless, please check the general response where we report the results on the inductive link prediction case where an inference graph is a totally new one comprised of new unseen entities - there we compare to most available works.
>
> Indeed, some analyses had to be moved to the appendix due to the lack of space. Generally, we did not observe significant differences when applying deterministic or random anchor sampling strategies on smaller graphs (including YAGO 3-10 of 120K nodes) although the deterministic strategy has more theoretical benefits as to the anchor distances distribution. For relations we find (as shown in the main ablations) that the key performance indicator is their general availability (relational context is there or not) despite the way they were obtained.

---

> > ### Comment · Reviewer_KuBz · 2021-11-27
> > **Reply to author response**
> >
> > Thank you for your response, as well as the inclusion of additional results for inductive link prediction. I believe your comments address my concerns, and I am happy to maintain my score.

---

### Author Response · Authors · 2021-11-12
**General Response to Reviewers**

We thank the reviewers for the constructive feedback. We appreciate that the reviewers highlight the merits of the paper as “clearly written, well motivated” (**KuBz**), “novel approach, will be much appreciated by practitioners” (**X7aq**), “especially useful in inductive learning” (**CpaB**) and “a promising direction” (**2qcD**). We further appreciate the comments on our experimental protocol finding that “the diverse range of experiments is a strong point of the work” (**KuBz**) and “empirical results strongly support the usefulness of the approach” (**X7aq**).

With NodePiece, we aim to bridge the gap between transductive and inductive setups by proposing a general representation learning framework freed of the limitations of shallow embedding tables (that are transductive-only & parameter inefficient), capable of obtaining representations of seen and unseen nodes (inductive) and do that for any node on-the-fly in the compositional manner. NodePiece features can be used with any GNN encoder and scoring function from KG embedding literature.

We would like to address a general concern on the breadth of KG representation learning benchmarks and baselines used in our empirical study. To this end, we add a set of experiments on the inductive link prediction benchmark [1] to measure the performance of NodePiece features in the case when anchor nodes are not available and only relational context can be used to compose entity representations. We plan to replace the Relation Prediction experiment with this new inductive LP experiment in the camera-ready version.

The unique feature of this benchmark compared to other evaluated tasks is that training and inference graphs are disjoint, i.e., inference at validation and test time is performed on a completely new graph comprised of new entities, and link prediction involves only entities unseen during training.

---

> ### Author Response · Authors · 2021-11-16
> **General Response Part II**
>
>
> Nodes in the inference graphs do not have any associated feature vectors which makes this benchmark very relevant for graph representation learning.
> Importantly, the set of relation types in the inference graphs is a subset of those seen in the training set.
>
> As inference graphs are disconnected from training ones, learning anchors based on the training graph is useless, so node hashes can be built only using the $m$-sized relational context.
>
> We compare NodePiece to strong baselines in path-based methods and GNNs including a recently proposed Neural Bellman-Ford framework [2]. In the revised version, we added Section I in the Appendix with full dataset description and evaluation protocol, here we report the Hits@10 metric on 12 splits (pertaining to subsets of FB15k-237, WN18RR and NELL-995) of various graph sizes.
>
> For each KG, there are 4 splits of increasing size of train and inference nodes and edges.
> The empirical results on this spectrum of various sizes demonstrate interesting scalability properties of NodePiece in inductive settings. Without anchor nodes, the NodePiece vocabulary size is independent of the number of nodes and edges, depending only on the number of learned relation types. On dense relation-rich graphs such a vocabulary is enough to yield very competitive prediction performance.
>
> |  | Method | FB V1 | FB V2 | FB V3 | FB V4 | WN V1 | WN V2 | WN V3 | WN V4 | NELL V1 | NELL V2 | NELL V3 | NELL V4 |
> |---|---|---|---|---|---|---|---|---|---|---|---|---|---|
> |  | Neural LP | 0.529 | 0.589 | 0.529 | 0.559 | 0.744 | 0.689 | 0.462 | 0.671 | 0.408 | 0.787 | 0.827 | 0.806 |
> |  | DRUM | 0.529 | 0.587 | 0.529 | 0.559 | 0.744 | 0.689 | 0.462 | 0.671 | 0.194 | 0.786 | 0.827 | 0.806 |
> |  | RuleN | 0.498 | 0.778 | 0.877 | 0.856 | 0.809 | 0.782 | 0.534 | 0.716 | 0.535 | 0.818 | 0.773 | 0.614 |
> |  | GraIL | 0.642 | 0.818 | 0.828 | 0.893 | 0.825 | 0.787 | 0.584 | 0.734 | 0.595 | **0.933** | 0.914 | 0.732 |
> |  | NBFNet | 0.692 | 0.858 | 0.898 | 0.923 | **0.942** | **0.895** | **0.900** | **0.881** | - | - | - | - |
> |  | NodePiece + CompGCN | **0.873** | **0.939** | **0.944** | **0.949** | 0.830 | 0.886 | 0.785 | 0.807 | **0.890** | 0.901 | **0.936** | **0.893** |
>
> Generally, the results confirm the trend identified in the node classification experiments: relation-only features are strong performers in dense relation-rich graphs.
> NodePiece features paired with CompGCN significantly improve over path-based methods where performance gap might reach 37 absolute Hits@10 points, e.g., in FB15k-237 V1 and NELL-995 V1.
>
>
> Comparing to GNNs, NodePiece + CompGCN outperforms GraIL by a large margin in all (except one) experiments and even outperforms Neural Bellman-Ford Nets, a recently proposed strong baseline for transductive and inductive link prediction, on relation-rich FB15k-237 splits.
> As expected, NodePiece features are less efficient on sparse graphs like WN18RR with few unique relations but still outperform GraIL.
>
> For each KG, there are 4 splits of increasing size of train and inference nodes and edges.
> The empirical results on this spectrum of various sizes demonstrates interesting scalability properties of NodePiece in inductive settings. Without anchor nodes, the NodePiece vocabulary size is independent of the number of nodes and edges, depending only on the number of learned relation types. On dense relation-rich graphs such a vocabulary is enough to yield very competitive prediction performance.
>
>
> [1] Teru et al. Inductive Relation Prediction by Subgraph Reasoning. ICML’20
> [2] Zhu et Neural Bellman-Ford Networks: A General Graph Neural Network Framework for Link Prediction. NeurIPS 2021

---

### Author Response · Authors · 2021-11-25
**Discussion Period**

Dear Reviewers,

we thank you for the valuable feedback that helped us to better position the work and support several claims with more experimental evidence which we put in the responses and the revised version of the manuscript. Please let us know if our clarifications and revisions addressed your concerns.

---

### Decision · Program_Chairs · 2022-01-20

**Decision:**

Accept (Poster)

**Comment:**

This paper presents a technique for compositionally constructing embeddings for nodes in knowledge graphs, hence reducing the memory requirements as well as allowing inductive learning. The reviewers find the direction promising and the approach novel and well-motivated. There were some concerns about the experiment results — Reviewer KuBz suggests including more baselines, Reviewer CpaB suggests trying NodePiece on single-relation graphs and Reviewer 2qcD notes that NodePiece lags behind the other approaches on some tasks. Most of these concerns seem to have been addressed in the author response and I tend to agree with the authors that single-relation graphs are out of the scope of this work. Reviewer X7aq also raised a concern about the claims made regarding (i) uniqueness of the hashes and (ii) sub-linearity of the approach. It is good to see that claim (ii) has been removed, but (i) is still present in many places — it would be good to add a discussion about why the hashes are highly likely to be unique in the final version.